# NHR-49/PPAR-α and HLH-30/TFEB cooperate for *C. elegans* host defense via a flavin-containing monooxygenase

Khursheed A Wani[1], Debanjan Goswamy[1], Stefan Taubert[2], Ramesh Ratnappan[3,4,5,6], Arjumand Ghazi[3,4,5,6], Javier E Irazoqui[1]*

[1]Department of Microbiology and Physiological Systems, UMass Medical School, Worcester, United States; [2]Department of Medical Genetics, University of British Columbia, Vancouver, Canada; [3]Department of Pediatrics, University of Pittsburgh School of Medicine, Pittsburgh, United States; [4]Department of Developmental Biology, University of Pittsburgh School of Medicine, Pittsburgh, United States ; [5]Department of Cell Biology, University of Pittsburgh School of Medicine, Pittsburgh, United States; [6]Department of Physiology, University of Pittsburgh School of Medicine, Pittsburgh, United States

**\*For correspondence:**
Javier.Irazoqui@umassmed.edu

**Competing interests:** The authors declare that no competing interests exist.

**Abstract** The model organism *Caenorhabditis elegans* mounts transcriptional defense responses against intestinal bacterial infections that elicit overlapping starvation and infection responses, the regulation of which is not well understood. Direct comparison of *C. elegans* that were starved or infected with *Staphylococcus aureus* revealed a large infection-specific transcriptional signature, which was almost completely abrogated by deletion of transcription factor *hlh-30/TFEB*, except for six genes including a flavin-containing monooxygenase (FMO) gene, *fmo-2/FMO5*. Deletion of *fmo-2/FMO5* severely compromised infection survival, thus identifying the first FMO with innate immunity functions in animals. Moreover, *fmo-2/FMO5* induction required the nuclear hormone receptor, NHR-49/PPAR-α, which controlled host defense cell non-autonomously. These findings reveal an infection-specific host response to *S. aureus*, identify HLH-30/TFEB as its main regulator, reveal FMOs as important innate immunity effectors in animals, and identify the mechanism of FMO regulation through NHR-49/PPAR-α during *S. aureus* infection, with implications for host defense and inflammation in higher organisms.

## Introduction

In their natural habitat, *C. elegans* feed on microbes that grow on rotting vegetable matter, and thus face a high likelihood of ingesting pathogens (*Schulenburg and Félix, 2017*). To defend against infection, *C. elegans* possess innate host defense mechanisms that promote their survival (*Ermolaeva and Schumacher, 2014*; *Kim and Ewbank, 2018*). In the laboratory, model human pathogenic bacteria cause intestinal pathology and death through poorly understood mechanisms (*Irazoqui et al., 2010a*). Infected animals experience both chemical signals that reveal the pathogen's presence and organismal stress caused by the infection. Over the last 15 years, several studies have identified and characterized *C. elegans* gene expression changes in response to pathogenic bacteria, fungi, and viruses, mounted through evolutionarily conserved mechanisms (*Irazoqui et al., 2010b*; *Kim and Ewbank, 2018*). However, the relative contributions of pathogen sensing and organismal stress mechanisms to the total pathogen-induced response remain unclear.

We previously showed that ingested Gram-positive bacterium *Staphylococcus aureus* causes drastic cytopathology in *C. elegans* (*Irazoqui et al., 2010a*). Infection with *S. aureus* results in progressive effacement and lysis of intestinal epithelial cells, whole-body cellular breakdown, and death

(*Irazoqui et al., 2010a*). Therefore, *S. aureus*-infected *C. elegans* experience dietary changes from its laboratory food of nonpathogenic *E. coli,* as well as intestinal destruction, cellular stress, and putative molecular signals produced by the pathogen.

In previous work, we showed that *C. elegans* mount a pathogen-specific transcriptional host response against *S. aureus,* which includes genes that encode antimicrobial proteins (e.g. lysozymes, antimicrobial peptides, and secreted C-type lectins) and cytoprotective factors (e.g. autophagy genes, lysosomal factors, and chaperones) that are necessary and sufficient for survival (*Irazoqui et al., 2010a*). Moreover, we discovered that *C. elegans* still induced select host defense genes even when exposed to heat-killed *S. aureus,* which did not cause intestinal destruction (*Irazoqui et al., 2010a*). However, the relative contributions of organismal stress and pathogen detection to the induction of the overall host defense response remain unknown.

We recently discovered that the induction of a large majority of the transcriptional host response to *S. aureus* requires HLH-30, the *C. elegans* homolog of mammalian transcription factor EB (TFEB) (*Visvikis et al., 2014*). TFEB belongs to the MiT family of transcription factors, which in mammals and *C. elegans* controls the transcription of autophagy and lysosomal genes in response to nutritional stress in addition to infection (*Lapierre et al., 2013*; *Raben and Puertollano, 2016*). HLH-30 and TFEB also regulate lipid store mobilization during nutritional deprivation (*O'Rourke and Ruvkun, 2013*; *Settembre et al., 2013*). Thus, HLH-30/TFEB could potentially integrate organismal stress, metabolism, and pathogen recognition to elicit coordinated host responses to infection. How HLH-30/TFEB integrates this information to produce stress-specific responses and what other factors are involved in such specificity are poorly understood. Specifically, the genes that are induced during infection independently of nutritional stress are not known.

Here, we report that *S. aureus* infection in *C. elegans* elicits a transcriptional response that is distinct from that induced by nutritional deprivation, thus defining an infection-specific transcriptional signature. Both the starvation response and the infection-specific signature were largely dependent on HLH-30/TFEB, highlighting its key role as a transcriptional integrator of organismal stress during infection. Moreover, we identified six genes that were specifically induced during infection even in the absence of HLH-30/TFEB, potentially revealing an alternative transcriptional host response signaling pathway. The induction of two of the six genes, *fmo-2/FMO5* and *K08C7.4,* was entirely dependent on transcription factor NHR-49/PPAR-α (*Van Gilst et al., 2005*), suggesting that NHR-49/PPAR-α defines an additional host defense pathway during *S. aureus* infection. NHR-49/PPAR-α was required non cell-autonomously for *fmo-2/FMO5* induction and host defense against *S. aureus*. Moreover, functional characterization of *fmo-2/FMO5* suggested that its enzymatic activity is specifically required for host defense against *S. aureus*, revealing that FMO-2/FMO5 is a key host defense effector. Thus, our work demonstrates for the first time that flavin-containing monooxygenases are important for host defense against infection in animals.

## Results

### Starvation and infection trigger distinct transcriptional responses

Our prior studies showed that *S. aureus* infection of *C. elegans* causes a robust host transcriptional response that results in the upregulation of 825 genes (*Irazoqui et al., 2010a*). Moreover, the 'early' phase of this response was already upregulated by 4 hr infection (*Irazoqui et al., 2010a*). It is likely that this transcriptional response to infection is compounded with nutritional stress, due to nutritional differences between laboratory food (nonpathogenic *E. coli*) and *S. aureus,* and due to intestinal destruction caused by the pathogen (*Irazoqui et al., 2010a*). To identify genes that are induced during infection independently of nutritional stress, we used whole-animal RNA-seq to directly compare infected and starved animals (*Figure 1A*). We identified 388 genes that were differentially expressed between these two conditions (*Figure 1B,C*, *Supplementary file 1*). About 70% (283) of differentially expressed genes were upregulated by starvation, while about 30% (105) were upregulated by infection (*Supplementary file 1*). Gene ontology (GO) analysis showed the starvation-induced genes to belong mostly to metabolic processes, whereas the infection-specific signature was highly enriched for innate immune response genes (*Supplementary file 2*). RT-qPCR of the 13 most highly infection-induced genes relative to animals that were starved or fed nonpathogenic *E. coli* laboratory food confirmed their *S. aureus*-specific induction (*Figure 1D*, *Figure 1—figure*

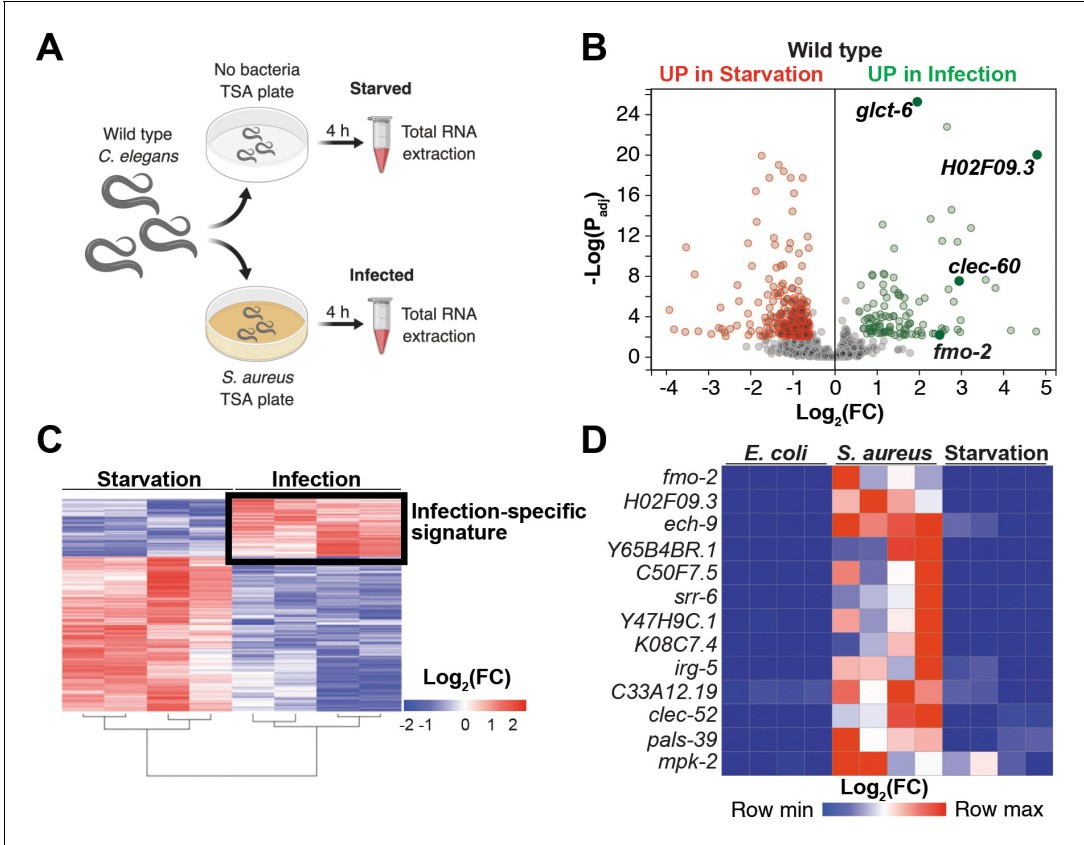

**Figure 1.** Starvation and *S. aureus* infection trigger distinct transcriptional responses. (**A**) Schematic overview of experimental approach for RNA-seq conditions. Synchronized young adults were subjected to either starvation or infection for 4 hr before RNA extraction. (**B**) Volcano plot of differentially expressed genes (P$_{adj}$ ≤ 0.01). Genes that were induced in each condition relative to the other are indicated in red (for starvation) and green (for infection). FC, fold change. P$_{adj}$, adjusted p value. (**C**) Heat map of differentially expressed genes [Log$_2$(FC)] comparing infection with *S. aureus* SH1000 to starvation by RNA-seq. The boxed area represents the designated infection-specific expression signature. (**D**) Heat map of a set of 13 genes most highly induced by *S. aureus* SH1000 compared to starvation, whose relative transcript levels were measured by RT-qPCR and plotted as row-normalized log$_2$(relative expression), or -ΔCt. Conditions include nonpathogenic *E. coli*, *S. aureus* (4 hr), and starvation (4 hr). Columns represent independent biological replicates.

The online version of this article includes the following source data and figure supplement(s) for figure 1:

**Figure supplement 1.** Expression analysis of 13 most highly induced genes.

**Figure supplement 1—source data 1.** mRNA levels of 13 highly induced genes in wild type animals fed nonpathogenic *E. coli*, infected with *S. aureus*, or starved.

**Figure supplement 1—source data 2.** mRNA levels of 13 highly induced genes in *hlh-30(-)* mutant animals fed nonpathogenic *E. coli*, infected with *S. aureus*, or starved.

---

supplement 1A). Thus, we identified an infection-specific signature of genes that excludes expression changes that are caused by starvation, indicating that the host responses to nutritional deprivation and *S. aureus* infection have distinct and specific features.

## HLH-30/TFEB is critical for host responses to starvation and infection

HLH-30/TFEB was shown to be important for gene induction during dietary challenge and during infection (*O'Rourke and Ruvkun, 2013*; *Settembre et al., 2013*; *Visvikis et al., 2014*). However, whether HLH-30/TFEB regulates the infection-specific response was not known. To assess the relevance of HLH-30/TFEB to the infection-specific signature, we compared starved and infected *hlh-30/ TFEB* loss-of-function mutants by RNA-seq. To our surprise, in *hlh-30/TFEB* mutants differential gene expression between starvation and infection was almost completely abrogated (*Figure 2A*, *Supplementary file 3*). Of the 105 genes in the infection-specific signature, only six were induced in *hlh-30/TFEB* mutants (*Figure 2B*), including *clec-52*, *fmo-2/FMO5*, and the uncharacterized genes

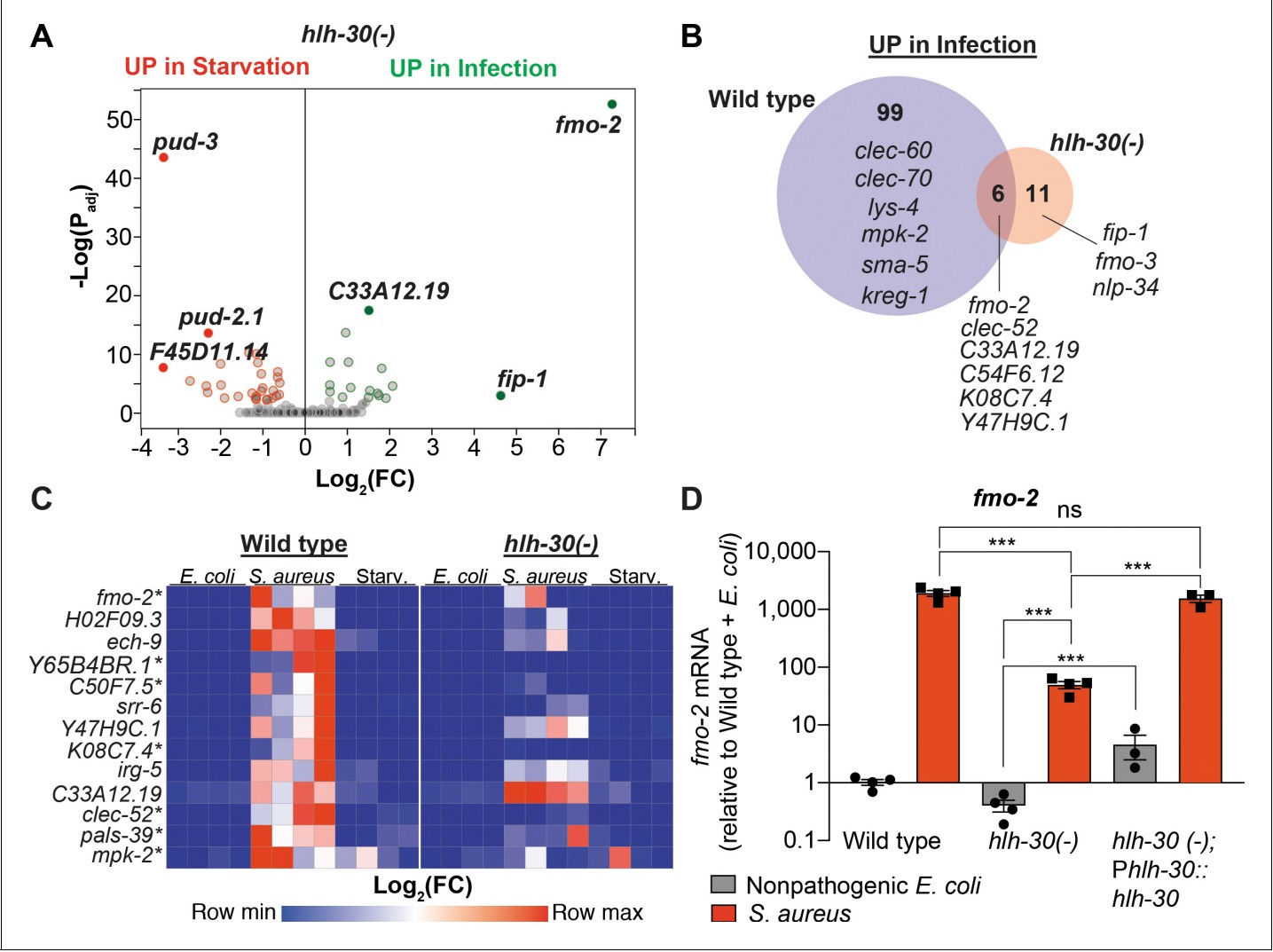

**Figure 2.** HLH-30/TFEB is critical for host responses to starvation and infection. (**A**) Volcano plot of differentially expressed genes in *hlh-30/TFEB* loss-of-function mutants ($P_{Adj.}$ ≤0.01). Genes that were induced in each condition relative to the other are indicated in red (for starvation) and green (for infection). (**B**) Venn diagram representing genes that were upregulated during infection compared to starvation in wild type and *hlh-30/TFEB* mutants. A few selected genes are indicated for reference. (**C**) Heat map of RT-qPCR (-ΔCt) relative expression values of a set of 13 genes most highly induced by *S. aureus* v starvation , measured in wild type and *hlh-30/TFEB* mutants. Conditions include nonpathogenic *E. coli*, *S. aureus*, and starvation. Columns represent independent biological replicates. * indicates genes that were highly induced in wild type compared to *hlh-30/TFEB* mutants during infection, and thus were partially or completely HLH-30/TFEB-dependent. 'Starv.', starvation. (**D**) RT-qPCR of *fmo-2/FMO5* transcript in wild type, *hlh-30/TFEB* loss-of-function mutants, and *hlh-30(-); Phlh-30::hlh-30::gfp* (complemented) animals fed nonpathogenic *E. coli* or infected with *S. aureus* (4 hr). Data are normalized to wild-type fed nonpathogenic *E. coli*, means ± SEM (3–4 independent biological replicates). ***p≤0.001, ns = not significant, one-way ANOVA followed by Šídák's test for multiple comparisons.

The online version of this article includes the following source data for figure 2:

**Source data 1.** *fmo-2* mRNA levels in wild type, *hlh-30(-)*, and *hlh-30(-); Phlh-30::hlh-30::gfp* (complemented) animals fed nonpathogenic *E. coli* or infected with *S. aureus*.

*C33A12.19*, *C54F6.12*, *K08C7.4*, and *Y47H9C.1* (**Supplementary file 3**). RT-qPCR confirmed the predicted results for the selected 13 top induced genes (**Figure 2C**, **Figure 1—figure supplement 1B**). Particularly, we verified that *fmo-2/FMO5* was partially induced in *hlh-30/TFEB* mutants compared to wild type (**Figure 2D**). Partial induction of *fmo-2/FMO5* in *hlh-30/TFEB* mutants was rescued by transgenic re-expression of *hlh-30/TFEB* driven by its endogenous promoter (**Figure 2D**). Altogether, these results showed that HLH-30/TFEB is crucial for both the starvation and the infection-

specific responses, and hinted at an HLH-30/TFEB-independent pathway for the induction of six infection-specific genes.

## Infection induces *fmo-2/FMO5* via NHR-49/PPAR-α

As the most highly induced infection-specific gene in *hlh-30/TFEB* mutants (*Figure 2A*, *Supplementary file 3*), *fmo-2/FMO5* attracted our attention. As shown previously (*Irazoqui et al., 2010a*) *fmo-2/FMO5* expression was induced several thousand-fold in *S. aureus*-infected animals relative to nonpathogenic *E. coli* controls (*Figure 3—figure supplement 1A*). In contrast, animals infected with Gram-negative pathogen *Pseudomonas aeruginosa* exhibited no significant change (*Figure 3—figure supplement 1A*), consistent with previous results (*Irazoqui et al., 2008*; *Irazoqui et al., 2010a*; *Wong et al., 2007*). The fluorescent *in vivo fmo-2/FMO5* transcriptional reporter showed faint GFP expression, mostly in the anterior intestine and head of noninfected animals (*Figure 3—figure supplement 1B*). Starvation modestly increased GFP expression in the intestine and nervous system (*Figure 3—figure supplement 1C*), while *P. aeruginosa* seemed to repress it below the levels observed in noninfected animals (*Figure 3—figure supplement 1D*) but upon quantification the difference was not significant (*Figure 3—figure supplement 1F*). In stark contrast, *S. aureus* caused high GFP induction in all tissues, except in gonads and eggs (*Figure 3—figure supplement 1E*). These observations confirmed that *fmo-2/FMO5* is strongly induced in a pathogen-specific manner. Our prior studies also showed that infection causes *fmo-2/FMO5* induction independently of previously identified host defense pathways, including p38 MAPK, TGF-β, ERK, insulin, Wnt, and HIF-1 pathways (*Irazoqui et al., 2008*; *Irazoqui et al., 2010a*; *Luhachack et al., 2012*; *Visvikis et al., 2014*). Additionally, we found that *fmo-2/FMO5* can be partially induced independently of HLH-30/TFEB (*Figure 2* and *Visvikis et al., 2014*). Therefore, additional transcriptional regulators must be involved in *fmo-2/FMO5* induction.

Previous studies identified NHR-49, a nuclear receptor homologous to human PPAR-α and HNF4-α, as essential for *fmo-2/FMO5* induction during exogenous oxidative stress (*Goh et al., 2018*; *Hu et al., 2018*). To examine the role of NHR-49/PPAR-α during *S. aureus* infection, we measured *fmo-2/FMO5* expression in *nhr-49/PPARA* null mutants (*Liu et al., 1999*; *Van Gilst et al., 2005*). We found that in these mutants, expression of the *fmo-2/FMO5* fluorescent transcriptional reporter was barely induced (*Figure 3F,G*) and, importantly, was undetectable in the intestinal epithelium (*Figure 3H–K*). In contrast, *fmo-2/FMO5* induction by *S. aureus* was partially dependent on *hlh-30/TFEB,* as predicted by RNA-seq (*Figure 3A–D and G*, *Figure 2*); in these mutants, expression was preserved in the pharyngeal isthmus, pharyngeal-intestinal valve, and in the intestinal epithelium, albeit to lower levels compared to wild type (*Figure 3H,I and K*). Thus, *nhr-49/PPARA* appeared to be essential for the induction of *fmo-2/FMO5* in the entire body. Consistently, noninfected *nhr-49/PPARA* mutants exhibited about 10-fold lower *fmo-2/FMO5* expression than wild type by RT-qPCR (*Figure 3L*). After infection, *nhr-49/PPARA* mutants completely failed to induce *fmo-2/FMO5* (*Figure 3L*). Transgenic rescue of *nhr-49/PPARA* driven by its endogenous promoter partially restored *fmo-2/FMO5* induction (*Figure 3L*). Therefore, *nhr-49/PPARA* was absolutely required for *fmo-2/FMO5* expression in infected and noninfected animals.

RT-qPCR of the other five HLH-30-independent genes showed that only *K08C7.4* induction by *S. aureus* was also dependent on NHR-49/PPAR-α (*Figure 3—figure supplement 2*). We generated a *K08C7.4* GFP transcriptional reporter strain and examined fluorescence in infected and noninfected wild type and *nhr-49/PPARA* mutants. In this strain, GFP was visible in noninfected animals in the anterior and posterior intestinal epithelium but was strongest in two unidentified head sensory neurons (as previously reported, potentially AFD neurons [*Lockhead et al., 2016*; *Mounsey et al., 2002*]), additional unidentified head and nerve ring neurons, ventral nerve cord, and an unidentified pair of tail neurons (*Figure 3—figure supplement 3A*). In infected animals, this expression pattern remained unchanged but increased slightly in intensity (*Figure 3—figure supplement 3B,E*). In contrast, *nhr-49/PPARA* mutants exhibited similar GFP expression in infected and noninfected animals (*Figure 3—figure supplement 3C–E*). Together, these data showed that *nhr-49/PPARA* was absolutely required for *fmo-2/PPARA* expression and induction, while it was only partially required for induction of *K08C7.4*. For this reason, we designated *K08C7.4* as *nfds-1* for '<u>N</u>HR-<u>f</u>orty-nine-<u>d</u>ependent induction by <u>S</u>. *aureus,* member 1'; *nfds-1* appears to have homologs only in nematodes. These data also suggested that NHR-49/PPAR-α contributes to the induction of some of the HLH-30-

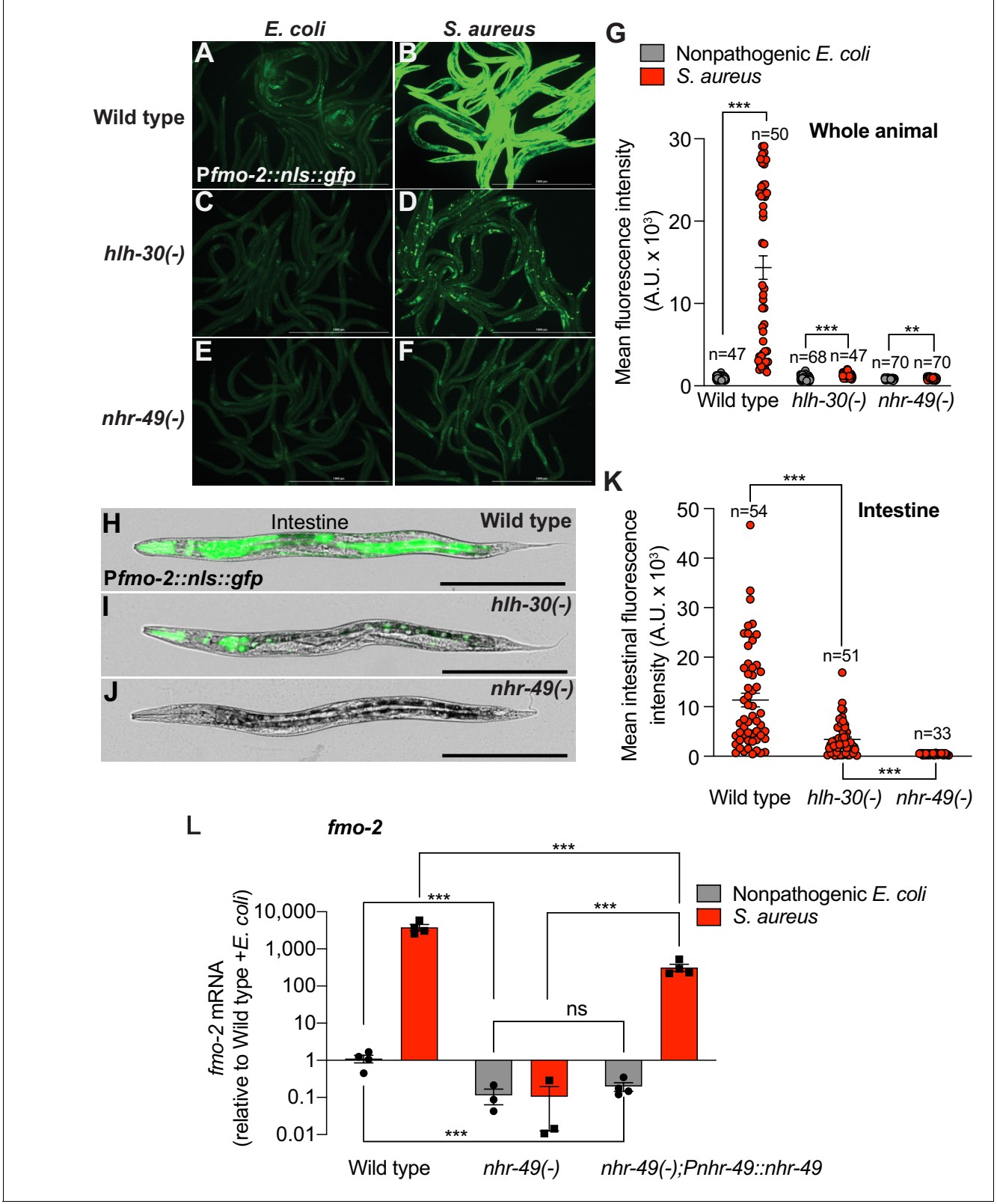

**Figure 3.** Infection induces *fmo-2/FMO5* via NHR-49/PPAR-α. (A–F) Epifluorescence micrographs of animals carrying P*fmo-2::nls::gfp* in wild type (A, B), *hlh-30/TFEB* (C, D), and *nhr-49/PPARA* mutant backgrounds (E, F) after feeding on *E. coli* OP50 or infection with *S. aureus* SH1000 (4 hr). Scale bar = 1000 μm. (G) Quantification of whole-animal P*fmo-2::nls::gfp* fluorescence in wild type, *hlh-30/TFEB*, and *nhr-49/PPARA* mutant animals after feeding on nonpathogenic *E. coli* or *S. aureus* (4 hr). Numbers atop bars indicate total number of animals in each condition. Error bars represent

*Figure 3 continued on next page*

Figure 3 continued

mean ± SEM. **p≤0.01, ***p≤0.001, unpaired two-sample two-tailed *t*-test. (H–J) High magnification epifluorescence images of P*fmo-2::nls::gfp* transgenic animals in wild type, *hlh-30(-)*, or *nhr-49(-)* mutant backgrounds after infection with *S. aureus* (4 hr). Scale bar = 300 μm. (K) Quantification of P*fmo-2::nls::gfp* fluorescence in the intestines of wild type, *hlh-30(-)*, and *nhr-49(-)* mutants after infection with *S. aureus* (4 hr). Numbers atop bars indicate total number of animals in each condition. Error bars represent mean ± SEM. ***p≤0.001, unpaired two-sample two-tailed *t*-test. (L) Relative expression of *fmo-2/FMO5* transcript (RT-qPCR -ΔCt) in wild type, *nhr-49(-)* mutants, and *nhr-49(-); Pnhr-49::nhr-49* (complemented) animals fed nonpathogenic *E. coli* OP50 or infected with *S. aureus* SH1000 (4 hr). Data are normalized to wild type on *E. coli,* means ± SEM (three to four independent biological replicates). ***p≤0.001, ns = not significant, one-way ANOVA followed by Šídák's test for multiple comparisons.

The online version of this article includes the following source data and figure supplement(s) for figure 3:

**Source data 1.** Quantification of whole-animal P*fmo-2::nls::gfp* fluorescence in wild type, *hlh-30(-)*, and *nhr-49(-)* animals after feeding on nonpathogenic *E. coli* or infected with *S. aureus*.

**Source data 2.** Quantification of P*fmo-2::nls::gfp* fluorescence in the intestines of wild type, *hlh-30(-)*, and *nhr-49(-)* mutants after infection with *S. aureus*.

**Source data 3.** *fmo-2* mRNA levels in wild type, *nhr-49(-)* mutants, and *nhr-49(-); Pnhr-49::nhr-49* (complemented) animals fed nonpathogenic *E. coli* or infected with *S. aureus*.

**Figure supplement 1.** *fmo-2/FMO5* is specifically and highly induced by *S. aureus*.

**Figure supplement 1—source data 1.** *fmo-2* mRNA levels in wild-type animals fed nonpathogenic *E. coli*, or infected with *S. aureus* or *P. aeruginosa*.

**Figure supplement 1—source data 2.** Quantification of P*fmo-2::nls::gfp* fluorescence in animals fed on *E. coli*, starved, or infected with *P. aeruginosa* or *S. aureus*.

**Figure supplement 2.** Expression analysis of HLH-30/TFEB-independent genes in *nhr-49/PPARA* mutants.

**Figure supplement 2—source data 1.** *clec-52* mRNA levels in wild type and *nhr-49(-)* mutant animals fed nonpathogenic *E. coli* or infected with *S. aureus*.

**Figure supplement 2—source data 2.** *Y47H9C.1* mRNA levels in wild type and *nhr-49(-)* mutant animals fed nonpathogenic *E. coli* or infected with *S. aureus*.

**Figure supplement 2—source data 3.** *K08C7.4* mRNA levels in wild type and *nhr-49(-)* mutant animals fed nonpathogenic *E. coli* or infected with *S. aureus*.

**Figure supplement 2—source data 4.** *C33A12.19* mRNA levels in wild type and *nhr-49(-)* mutant animals fed nonpathogenic *E. coli* or infected with *S. aureus*.

**Figure supplement 2—source data 5.** *C54F6.12* mRNA levels in wild type and *nhr-49(-)* mutant animals fed nonpathogenic *E. coli* or infected with *S. aureus*.

**Figure supplement 3.** *K08C7.4* induction requires NHR-49/PPAR-α.

**Figure supplement 3—source data 1.** Quantification of P*K08C7.4::gfp* fluorescence in wild type and *nhr-49(-)* mutant animals fed *E. coli* or infected with *S. aureus*.

independent host defense genes, but the biological significance of NHR-49/PPAR-α to host defense was not clear.

## NHR-49/PPAR-α is required for host defense

Compared to wild type, *nhr-49/PPARA* null mutants showed defective survival of *S. aureus* infection (*Figure 4A*) and shorter lifespan when fed nonpathogenic *E. coli* (*Figure 4B*), as previously reported (*Van Gilst et al., 2005*), suggesting that NHR-49/PPAR-α may have important roles in both host defense and aging. Transgenic rescue of *nhr-49/PPARA* driven by its endogenous promoter completely rescued the infection survival defect (*Figure 4A*) but only partially restored the total life-span on *E. coli* (*Figure 4B*), suggesting that distinct thresholds of NHR-49/PPAR-α function exist in infection and aging. Moreover, relative to wild type, two distinct *nhr-49/PPARA* gain-of-function mutants (*Lee et al., 2016*; *Svensk et al., 2013*) showed enhanced infection survival (*Figure 4C*). In contrast, gain-of-function mutant *nhr-49(et7)* (gf1) exhibited prolonged lifespan on *E. coli*, while *nhr-49(et8)* (gf2) exhibited shortened lifespan (*Figure 4D*), consistent with previous results (*Lee et al., 2016*). These results showed that NHR-49/PPAR-α promotes host infection survival, while its function in aging may be more complex.

Also in line with previous results (*Lee et al., 2016*), RT-qPCR of noninfected animals showed con-stitutively elevated *fmo-2/FMO5* expression in gain-of-function *nhr-49/PPARA* mutants relative to wild type (*Figure 4E*), consistent with the observed pro-survival function of NHR-49/PPAR-α. Upon infection, both gain-of-function mutants exhibited further *fmo-2/FMO5* induction, reaching higher *fmo-2/FMO5* expression than wild type controls (*Figure 4E*). Consistently, on nonpathogenic *E. coli* *nhr-49/PPARA* gain-of-function mutants exhibited constitutively high *fmo-2/FMO5* reporter GFP expression in the anterior pharynx, pharyngeal isthmus, nervous system, and the intestinal epithelium (*Figure 4F,H*, *Figure 4—figure supplement 1*). Infection further increased reporter expression

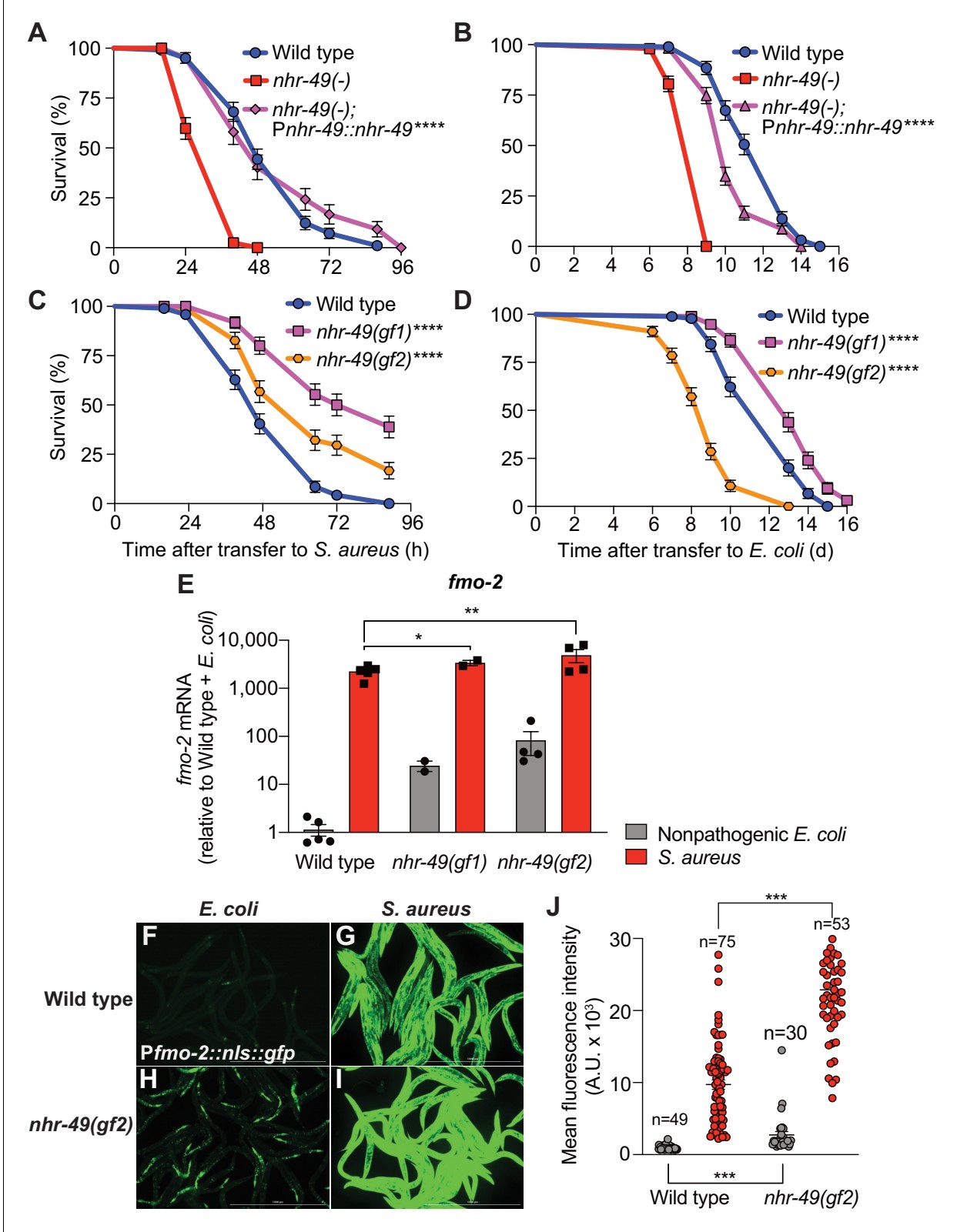

**Figure 4.** NHR-49/PPAR-α is required for host defense against infection. (**A**) Survival of wild type, *nhr-49/PPARA* loss-of-function, and *nhr-49(-)*; P*nhr-49::nhr-49* (complemented) animals infected with *S. aureus* SH1000. Data are representative of two independent trials. ****p≤0.0001 (Log-Rank test). (**B**) Lifespan on nonpathogenic *E. coli* OP50 of wild type, *nhr-49/PPARA* loss-of-function, and *nhr-49(-)*; P*nhr-49::nhr-49* animals. Data are representative of two independent trials. ****p≤0.0001 (Log-Rank test). (**C**) Survival of wild type and two *nhr-49/PPARA* gain-of-function mutants (gf1 = *et7* and gf2 = *et8*) *Figure 4 continued on next page*

*Figure 4 continued*

infected with *S. aureus* SH1000. Data are representative of two independent trials. ****p≤0.0001 (Log-Rank test). (D) Lifespan of wild type and *nhr-49/PPARA* gain-of-function mutants on *E. coli* OP50. Data are representative of three independent trials. ****p≤0.0001 (Log-Rank test). (E) Relative expression of *fmo-2/FMO5* transcript (RT-qPCR -ΔCt) in wild type and *nhr-49/PPARA* gain-of-function mutants fed nonpathogenic *E. coli* OP50 or infected with *S. aureus* SH1000 (4 hr). Data are normalized to wild type on *E. coli*, means ± SEM (two to five independent biological replicates). *p≤0.05, **p≤0.01, unpaired two-sample two-tailed *t*-test. (F–I) Epifluorescence micrographs of P*fmo-2::nls::gfp* in wild type (F–G) and *nhr-49(gf2)* mutants (H–I) fed nonpathogenic *E. coli* OP50 or infected with *S. aureus* SH1000 (4 hr). Scale bar = 1000 μm. (J) Quantification of GFP fluorescence in wild type and *nhr-49(gf2)* animals expressing P*fmo-2::nls::gfp*, after feeding on *E. coli* or infection with *S. aureus* (4 hr). Number of animals is indicated atop the bars. Error bars represent mean ± SEM. ***p≤0.001, unpaired two-sample two-tailed *t*-test.

The online version of this article includes the following source data and figure supplement(s) for figure 4:

**Source data 1.** Survival of wild type, *nhr-49(-)* mutant, and *nhr-49(-); Pnhr-49::nhr-49* (complemented) animals infected with *S. aureus*.
**Source data 2.** Lifespan of wild type, *nhr-49(-)* mutant, and *nhr-49(-); Pnhr-49::nhr-49* (complemented) animals on nonpathogenic *E. coli*.
**Source data 3.** Survival of wild type and *nhr-49* gain-of-function mutants (*et7* and *et8*) infected with *S. aureus*.
**Source data 4.** Lifespan of wild type and *nhr-49* gain-of-function mutants (*et7* and *et8*) on nonpathogenic *E. coli*.
**Source data 5.** *fmo-2* transcript levels in wild type and *nhr-49* gain-of-function mutants (*et7* and *et8*) fed nonpathogenic *E. coli* or infected with *S. aureus*.
**Source data 6.** Quantification of P*fmo-2::nls::gfp* fluorescence in wild type and *nhr-49(et8)* animals fed *E. coli* or infected with *S. aureus*.
**Figure supplement 1.** NHR-49/PPAR-α gain-of-function causes constitutive induction of *fmo-2*.

throughout the body in *nhr-49/PPARA* gain-of-function mutants, becoming much stronger than wild-type animals (*Figure 4G,I,J*, *Figure 4—figure supplement 1*). These results suggested that *nhr-49/PPARA* activation is sufficient for *fmo-2/FMO5* expression in a spatially restricted pattern, and confirmed that infection synergistically upregulates *fmo-2/FMO5* expression throughout the entire body. Interestingly, the pattern of *fmo-2/FMO5* expression in noninfected *nhr-49/PPARA* gain-of-function mutants (*Figure 4H*, *Figure 4—figure supplement 1*) resembled that observed in infected *hlh-30/TFEB* null mutants, where NHR-49/PPAR-α would be expected to be active (*Figure 3D,I*, *Figure 4—figure supplement 1*).

## NHR-49/PPAR-α functions in multiple tissues for host defense

*nhr-49/PPARA* is expressed in multiple tissues (*Ratnappan et al., 2014*). To identify specific tissues where *nhr-49/PPARA* is sufficient for host defense against *S. aureus*, we reintroduced wild type *nhr-49/PPARA* into *nhr-49/PPARA* mutants driven by tissue-specific promoters, including intestine, neurons, muscle, and epidermis (i.e. hypodermis). We examined these rescue lines for *fmo-2/FMO5* induction and survival of infection. Intestinal rescue of *nhr-49/PPARA* fully restored both basal and induced *fmo-2/FMO5* expression (*Figure 5A*). Re-expression of *nhr-49/PPARA* partially restored *fmo-2/FMO5* induction in other lines (*Figure 5D,G,J*). Consistently, expression in each of these tissues also rescued the infection survival defect of *nhr-49/PPARA* mutants. In fact, intestinal, neuronal, and muscular expression enhanced infection survival compared to wild type (*Figure 5B,E,H*), while epidermal expression rescued infection survival to a level similar to wild type (*Figure 5K*). These results suggested that *nhr-49/PPARA* can function from any of these tissues to promote host defense against infection.

In contrast, tissue-specific complementation of *nhr-49/PPARA* had more complex effects on normal lifespan on nonpathogenic *E. coli*. Intestinal and epidermal re-expression prolonged lifespan compared to wild type (*Figure 5C,L*). Neuronal expression rescued lifespan to the wild-type level (*Figure 5F*), and muscle expression yielded partial rescue (*Figure 5I*). Together, these data showed distinct and tissue-specific roles for *nhr-49/PPARA* in host defense and lifespan.

## NHR-49/PPAR-α controls a fraction of the infection-specific transcriptional signature

To better understand the biological relevance of *nhr-49/PPARA* to the infection-specific host response, we compared the transcriptomes of starved and infected *nhr-49/PPARA* mutants. In stark contrast to *hlh-30/TFEB* mutants, which showed a much-reduced differential response compared to wild type (*Figure 2*), *nhr-49/PPARA* mutants exhibited many more differentially expressed genes than wild type between these two conditions (*e.g.* 313 v. 135 upregulated, *Figure 6* and *Supplementary file 4*). Moreover, 92 (68%) of the 135 infection-upregulated genes in wild type were also upregulated in *nhr-49/PPARA* mutants, and we categorized them as NHR-49/PPAR-α-

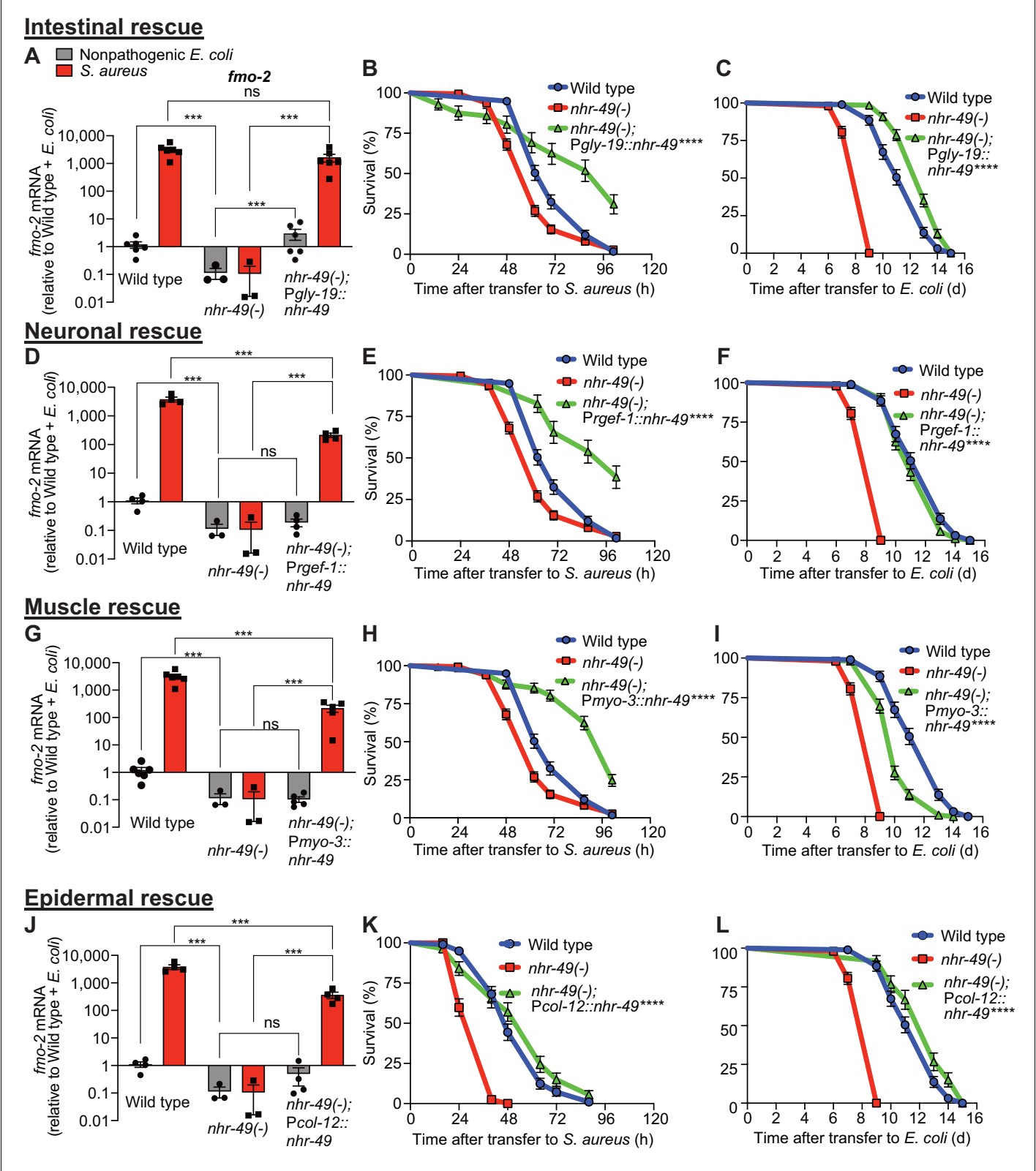

**Figure 5.** NHR-49/PPAR-α functions in multiple tissues for host defense. (A, D, G, J) Relative expression of *fmo-2/FMO5* transcript (RT-qPCR -ΔCt) in wild type, *nhr-49/PPARA* loss-of-function mutants, and tissue-specific *nhr-49/PPARA* rescue lines: P*glp-19* for intestine, (P*glp-19::nhr-49::gfp*), P*rgef-1* for nervous system (P*rgef-1::nhr-49::gfp*), P*myo-3* for body wall muscle (P*myo-3::nhr-49::gfp*), and P*col-12* for epidermis (P*col-12::nhr-49::gfp*); fed nonpathogenic *E. coli* OP50 or infected with *S. aureus* SH1000 (4 hr). Data are normalized to wild type on *E. coli*, means ± SEM (three to six

*Figure 5 continued on next page*

Figure 5 continued

independent biological replicates,). ***p≤0.001, ns = not significant, one-way ANOVA followed by Šídák's test for multiple comparisons. (B, E, H, K) Survival of wild type, *nhr-49/PPARA* loss of function, and tissue-specific *nhr-49/PPARA* rescue lines infected with *S. aureus*. Data are representative of two independent trials. ****p≤0.0001 (Log-Rank test). Comparisons are made between *nhr-49(-)* and the rescue lines. (C, F, I, L) Lifespan of wild type, *nhr-49/PPARA* loss of function, and tissue-specific *nhr-49* rescue lines on nonpathogenic *E. coli*. Data are representative of three independent trials. ****p≤0.0001 (Log-Rank test). Comparisons are made between *nhr-49(-)* and the rescue lines.

The online version of this article includes the following source data for figure 5:

**Source data 1.** *fmo-2* mRNA levels in wild type, *nhr-49(-)* mutant, and intestine-specific (using *gly-19* promoter) *nhr-49* rescue line animals.
**Source data 2.** Survival of wild type, *nhr-49(-)* mutant, and intestine-specific (using *gly-19* promoter) *nhr-49* rescue line animals infected with *S. aureus*.
**Source data 3.** Lifespan of wild type, *nhr-49(-)* mutant, and intestine-specific (using *gly-19* promoter) *nhr-49* rescue line animals on nonpathogenic *E. coli*.
**Source data 4.** *fmo-2* mRNA levels in wild type, *nhr-49(-)* mutant, and neuron-specific (using *rgef-1* promoter) *nhr-49* rescue line animals.
**Source data 5.** Survival of wild type, *nhr-49(-)* mutant, and neuron-specific (using *rgef-1* promoter) *nhr-49* rescue line animals infected with *S. aureus*.
**Source data 6.** Lifespan of wild type, *nhr-49(-)* mutant, and neuron-specific (using *rgef-1* promoter) *nhr-49* rescue line animals on nonpathogenic *E. coli*.
**Source data 7.** *fmo-2* mRNA levels in wild type, *nhr-49(-)* mutant, and muscle-specific (using *myo-3* promoter) *nhr-49* rescue line animals.
**Source data 8.** Survival of wild type, *nhr-49(-)* mutant, and muscle-specific (using *myo-3* promoter) *nhr-49* rescue line animals infected with *S. aureus*.
**Source data 9.** Lifespan of wild type, *nhr-49(-)* mutant, and muscle-specific (using *myo-3* promoter) *nhr-49* rescue line animals on nonpathogenic *E. coli*.
**Source data 10.** *fmo-2* mRNA levels in wild type, *nhr-49(-)* mutant, and epidermis-specific (using *col-12* promoter) *nhr-49* rescue line animals.
**Source data 11.** Survival of wild type, *nhr-49(-)* mutant, and epidermis-specific (using *col-12* promoter) *nhr-49* rescue line animals infected with *S. aureus*.
**Source data 12.** Lifespan of wild type, *nhr-49(-)* mutant, and epidermis-specific (using *col-12* promoter) *nhr-49* rescue line animals on nonpathogenic *E. coli*.

independent (*Figure 6C* and *Supplementary file 4*). GO analysis detected overrepresentation of molecular function categories related to *Lysozyme Activity* (GO:0003796, $P_{adj}$ = 0.033808778, genes: *ilys-2, ilys-3,* and *lys-3*), and of biological process categories related to *Innate Immune Response* (GO:0045087, $P_{adj}$ = 6.62E-16, genes including *clec-86, dct-17, dod-19,* and *tsp-1*). Although KEGG pathway analysis did not detect overrepresentation of any known pathways, analysis of TRANSFAC motifs identified great overrepresentation of aryl hydrocarbon receptor (AHR) motif (M00000, $P_{adj}$ = 1.11E-26) and of the consensus motif for ELT-3 (GATA2, M07154, $P_{adj}$ = 1.20E-10). These data imply that in wild type animals, the NHR-49-independent component of the infection-specific signature is enriched for antimicrobial enzymes and innate immune response genes, potentially driven by AHR and ELT-3.

Only 43 genes were induced by infection in wild type but not in *nhr-49/PPARA* mutants, and thus we categorized them as NHR-49-dependent (*Figure 6C* and *Supplementary file 4*). GO analysis of these genes detected over-representation of molecular function categories related to *Monooxygenase Activity* (GO:0004497, $P_{adj}$ = 0.000157209), *NAD(P)H Oxidase $H_2O_2$-forming Activity* (GO:0016174, $P_{adj}$ = 0.002332572), and *Carbohydrate Binding* (GO:0030246, $P_{adj}$ = 0.015473897). The first two relate to flavin-containing monooxygenase genes *fmo-2* and *fmo-5*, in addition to *drd-1* (FAXDC2 homolog) and *cyp-14A3* (Cytochrome P450). The last category relates to the expression of C-type Lectin (CLEC) genes *clec-52, clec-172, clec-204,* and *clec-265*. Additionally, the most relevant overrepresented biological process categories were related to *Lipid Metabolic Process* (GO:0006629, $P_{adj}$ = 4.29265E-06, genes: *Y73B6BL.37, Y65B4BR.1, Y46H3A.5, C39B5.14, pmp-1, drd-1, K05B2.4,* and *sptl-2*, which is the entry point to the ceramide and sphingosine biosynthetic pathways) and *Defense Response to Gram-Positive Bacterium* (GO:0050830, $P_{adj}$ = 0.003214175). Examples included C-type lectin *clec-60*, lysozyme *lys-5*, *fmo-2/FMO5*, and *K08C7.4*. KEGG pathway analysis also detected overrepresentation of Cytochrome P450 Drug Metabolism (KEGG:00982), related to genes *fmo-2, fmo-5,* and *gst-5*. Analysis of TRANSFAC motifs also identified strong over-representation of the AHR motif (M00000, $P_{adj}$ = 4.59E-15) in this gene set. Therefore, we concluded that in wild-type animals, NHR-49/PPAR-α is required for the expression of redox enzymes, C-type lectins, and lipid metabolic enzymes, possibly involving AHR.

In contrast, 221 genes were induced by infection only in *nhr-49/PPARA* mutants, and thus it appeared that *nhr-49/PPARA* loss may be compensated by a large infection-specific response that does not normally occur in wild-type animals. By GO analysis, this group was enriched for one molecular function category related to *Carbohydrate Binding* (GO:0030246, $P_{adj}$ = 3.86602E-05), by genes *clec-70, clec-71, clec-9, clec-118, clec-125, clec-186, clec-187, clec-221,* and *Y38H6C.8* (orthologous to mammalian antimicrobial C-type lectin HIP/PAP/REG3γ), galectin gene *lec-11*, and α-mannosidase gene *aman-3*. The most relevant overrepresented biological process categories were

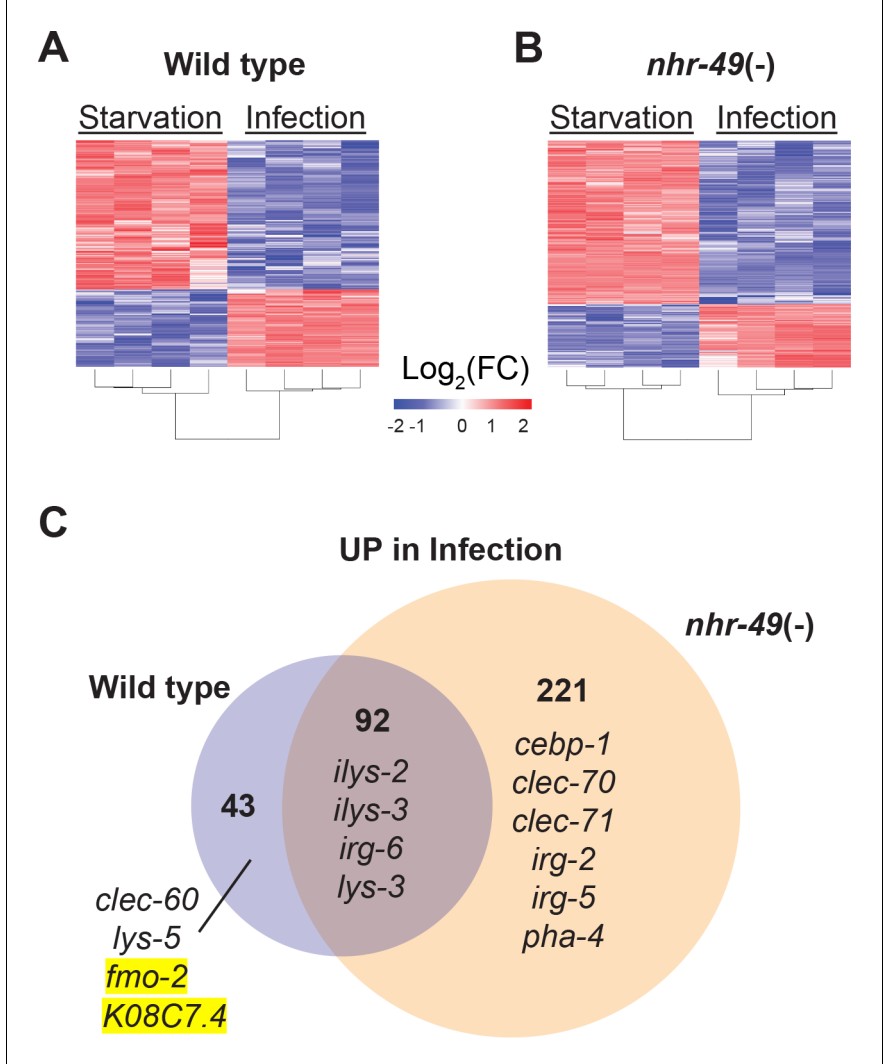

**Figure 6.** NHR-49/PPAR-α is required for one-third of the infection-specific transcriptional signature. (**A–B**) Heat map of differentially expressed genes during starvation and infection in wild type (**A**) and *nhr-49(-)* (**B**) animals (RNA-seq, Log₂(FC), $P_{Adj}$ ≤0.001). Columns represent a biological replicate each. (**C**) Venn diagram representing genes that are upregulated by 4 hr *S. aureus* infection compared with 4 hr starvation in wild type and *nhr-49(-)* animals. Shown are a few examples for reference.

related to *Response to External Biotic Stimulus* (GO:0043207, $P_{adj}$ = 8.41E-33) and *Innate Immune Response* (GO:0045087, $P_{adj}$ = 2.15E-26), suggesting that in the absence of NHR-49/PPAR-α, infection induces an alternative innate immune response. Remarkably, KEGG pathway analysis did not detect overrepresentation of any known pathways, indicating that the compensatory response does not enhance any particular aspect of metabolism or signaling that can be detected by transcriptional analysis. TRANSFAC motif analysis again detected strong enrichment of the AHR motif (M00000, $P_{adj}$ = 3.80E-51) and of the ELT-3 motif (M07154, $P_{adj}$ = 9.65E-21), suggesting that AHR may have a broad role in the overall infection-specific signature, while ELT-3 has a more restricted role in the NHR-49-independent and compensatory signatures. TRANSFAC analysis also detected overrepresentation of the SKN-1/NRF2 motif (M00230, $P_{adj}$ = 0.007316808), suggesting that in the absence of NHR-49/PPAR-α, SKN-1/NRF2 may become activated during *S. aureus* infection.

## HLH-30/TFEB genetically functions downstream of NHR-49/PPAR-α for host defense

During infection, *nhr-49/PPARA* expression did not change in wild-type animals compared to uninfected controls (*Figure 7A*). Moreover, *nhr-49/PPARA* expression was similar in noninfected wild type and *hlh-30/TFEB* mutants. In contrast, in infected *hlh-30/TFEB* mutants compared with wild type, *nhr-49/PPARA* expression was lower (*Figure 7A*), indicating that HLH-30/TFEB contributes to *nhr-49/PPARA* expression during infection. Conversely, in *nhr-49/PPARA* null mutants *hlh-30/TFEB* baseline expression was higher than in wild type, yet its induction by infection was abrogated (*Figure 7B*). This indicated that *nhr-49/PPARA* is required for increased expression of *hlh-30/TFEB* during infection. Moreover, in *nhr-49/PPARA* gain-of-function mutants compared to wild type, both *hlh-30/TFEB* baseline expression and induction were higher (*Figure 7B*). Considered together, these data suggested that HLH-30/TFEB and NHR-49/PPAR-α contribute to each other's expression in noninfected and infected animals in different ways.

Because *hlh-30/TFEB* and *nhr-49/PPARA* contributed to both infection-specific *fmo-2/FMO5* induction (*Figure 2*, *Figure 3*) and each other's expression (*Figure 7A–B*), we examined their genetic interactions. To determine whether *hlh-30/TFEB* and *nhr-49/PPARA* genetically function in the same pathway, we first knocked down *hlh-30/TFEB* in wild type and *nhr-49PPARA* null mutants by RNA interference (RNAi). Whereas *hlh-30/TFEB* knockdown greatly impaired *S. aureus* infection survival in wild type animals, it did not affect *nhr-49/PPARA* mutants (*Figure 7—figure supplement 1A*). In contrast, *hlh-30/TFEB* knockdown shortened total lifespan in both wild type and *nhr-49/PPARA* mutants fedon nonpathogenic *E. coli* (*Figure 7—figure supplement 1B*). These results are consistent with a model in which *hlh-30*/TFEB and *nhr-49/PPARA* function in parallel pathways for longevity, while they function in the same pathway for infection survival.

To test the proposed model, we constructed *nhr-49/PPARA* (gain-of-function); *hlh-30/TFEB* (loss-of-function) double mutants. Remarkably, despite enhancing infection survival as single mutants (*Figure 4C*), in combination with *hlh-30(-)* neither *nhr-49*(gf1) nor *nhr-49*(gf2) affected host survival (*Figure 7C*). Therefore, *hlh-30/TFEB* deletion completely suppressed the enhanced survival phenotypes of *nhr-49/PPARA* gain-of-function alleles, which is consistent with *hlh-30/TFEB* functioning downstream of *nhr-49/PPARA* for host infection survival. In contrast, both double mutants showed increased lifespan on nonpathogenic *E. coli* compared to *hlh-30/TFEB* single mutants (*Figure 7D*), suggesting that *nhr-49PPARA* functions downstream of, or parallel to, *hlh-30/TFEB* for lifespan determination.

Analysis of *fmo-2/FMO5* expression in the double mutants showed that *hlh-30/TFEB* was required for full induction by *nhr-49/PPARA* gain of function in both infected and noninfected animals (*Figure 7E*). Because *hlh-30/TFEB* mutation did not fully suppress either gain-of-function allele, for the *fmo-2/FMO5* expression phenotype *hlh-30/TFEB* and *nhr-49/PPARA* appear to function in parallel (additive) pathways and thus, *hlh-30/TFEB* is only partially required for *fmo-2/FMO5* expression.

## FMO-2/FMO5 is required for host survival of infection

So far, we focused on *fmo-2/FMO5* as a useful reporter of the broader infection-specific signature, but its biological relevance to infection survival was unknown. FMO-2 and FMO5 belong to the evolutionarily conserved flavin-containing monooxygenase (FMO) protein family (*Huijbers et al., 2014*). In mammals, FMO proteins are primarily known to function in the detoxification of foreign substances (xenobiotics) with prominent roles in drug metabolism (*Krueger and Williams, 2005*). *C. elegans* FMO-2 exhibits homology to human proteins FMO1-5, with closest similarity to FMO5 (42% identity). Previously, FMO-2/FMO5 had been implicated in dietary-restriction-mediated lifespan extension, and its forced expression resulted in stress resistance (*Leiser et al., 2015*). In plants, FMOs participate in host defense against bacterial and fungal infections (*Bartsch et al., 2006*; *Koch et al., 2006*). Whether animal FMOs also function in innate host defense was not known.

To determine the physiological relevance of *fmo-2/FMO5* during infection, we examined mutants homozygous for an *fmo-2/FMO5* deletion predicted to result in a null allele (*C. elegans Deletion Mutant Consortium, 2012*). Compared with wild type, *fmo-2/FMO5* mutants exhibited greatly compromised survival of *S. aureus* infection (*Figure 8A*) but did not exhibit differences in survival of *P. aeruginosa* infection or in aging when fed nonpathogenic *E. coli* (*Figure 8B–C*). Deletion of *fmo-2/FMO5* did not affect the induction of the nine most highly induced infection-specific signature genes

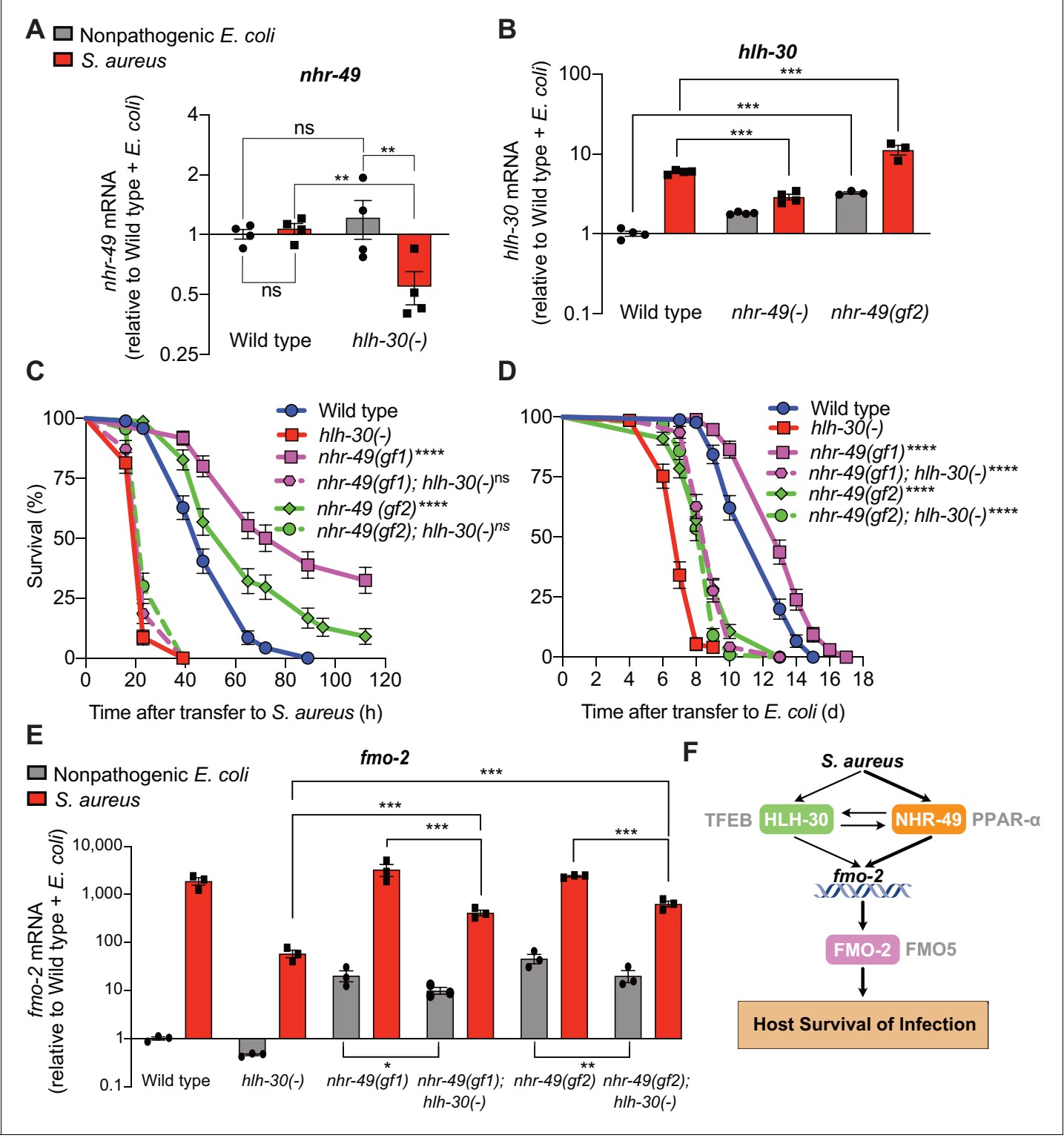

**Figure 7.** HLH-30/TFEB genetically functions downstream of NHR-49/PPAR-α for host defense. (**A**) Relative expression of *nhr-49/PPARA* transcript (RT-qPCR -ΔCt) in wild type and *hlh-30/TFEB* loss-of-function mutants fed nonpathogenic *E. coli* OP50 or infected with *S. aureus* SH1000 (8 hr). Data are normalized to wild type on *E. coli,* means ± SEM (four independent biological replicates). **p≤0.01, ns = not significant, one-way ANOVA followed by Šídák's test for multiple comparisons. (**B**) Relative expression of *hlh-30/TFEB* transcript (RT-qPCR -ΔCt) in wild type, *nhr-49/PPARA* loss of function, and *nhr-49/PPARA* gain-of-function (gf2) mutants fed nonpathogenic *E. coli* OP50 or infected with *S. aureus* SH1000 (8 hr). Data are normalized to wild type on *E. coli,* means ± SEM (three to four independent biological replicates ). ***p≤0.001, one-way ANOVA followed by Šídák's test for multiple

*Figure 7 continued on next page*

Figure 7 continued

comparisons. (C) Survival of wild type, *hlh-30/TFEB* loss of function, *nhr-49(gf1)*, *nhr-49(gf2)*, *nhr-49(gf1); hlh-30(-)*, and *nhr-49(gf2); hlh-30(-)* animals infected with *S. aureus* SH1000. Data are representative of two independent trials. ****p≤0.0001 (Log-Rank test, compared to *hlh-30*(-) mutants). (D) Lifespan of wild type, *hlh-30/TFEB* loss of function, *nhr-49(gf1)*, *nhr-49(gf2)*, *nhr-49(gf1); hlh-30(-)*, and *nhr-49(gf2); hlh-30(-)* animals on nonpathogenic *E. coli* OP50. Data are representative of two independent trials. ****p≤0.0001 (Log-Rank test, compared to *hlh-30*(-) mutants). (E) Relative expression of *fmo-2/FMO5* transcript (RT-qPCR -ΔCt) in wild type, *hlh-30(-)*, *nhr-49(gf1)*, *nhr-49(gf1); hlh-30(-)*, *nhr-49(gf2)*, and *nhr-49(gf2); hlh-30(-)* animals fed nonpathogenic *E. coli* OP50 or infected with *S. aureus* SH1000 (4 hr). Data are normalized to wild type on *E. coli*, means ± SEM (three to four independent biological replicates). *p≤0.05, **p≤0.01, ***p≤0.001, one-way ANOVA followed by Šídák's test for multiple comparisons. (F) Schematic representation of *fmo-2/FMO5* regulation during infection with *S. aureus*. Human homologs of the *C. elegans* proteins are indicated in grey lettering.

The online version of this article includes the following source data and figure supplement(s) for figure 7:

**Source data 1.** *nhr-49* mRNA levels in wild type and *hlh-30(-)* mutant animals fed nonpathogenic *E. coli* or infected with *S. aureus*.
**Source data 2.** *hlh-30* mRNA levels in wild type, *nhr-49(-)*, and *nhr-49* gain-of-function (gf2) mutants fed nonpathogenic *E. coli* or infected with *S. aureus*.
**Source data 3.** Survival of wild type, *hlh-30(-)*, *nhr-49(gf1)*, *nhr-49(gf2)*, *nhr-49(gf1); hlh-30(-)*, and *nhr-49(gf2); hlh-30(-)* animals infected with *S. aureus*.
**Source data 4.** Lifespan of wild type, *hlh-30(-)*, *nhr-49(gf1)*, *nhr-49(gf2)*, *nhr-49(gf1);hlh-30(-)*, and *nhr-49(gf2);hlh-30(-)* animals on non-pathogenic *E. coli*.
**Source data 5.** *fmo-2* mRNA levels in wild type, *hlh-30(-)*, *nhr-49(gf1)*, *nhr-49(gf1);hlh-30(-)*, *nhr-49(gf2)*, and *nhr-49(gf2);hlh-30(-)* animals fed nonpathogenic *E. coli* or infected with *S. aureus*.
**Figure supplement 1.** HLH-30/TFEB functions downstream or parallel to NHR-49/PPAR-α.
**Figure supplement 1—source data 1.** Survival of wild type and *nhr-49(-)* mutant animals, infected with *S. aureus*, fed on either control RNAi, or RNAi against *hlh-30*.
**Figure supplement 1—source data 2.** Lifespan of wild type and *nhr-49(-)* mutant animals, on non-pathogenic *E. coli*, fed on either control RNAi, or RNAi against *hlh-30*.

(*Figure 8—figure supplement 1*). These data suggested that FMO-2/FMO5 may play an important role for host defense specifically during *S. aureus* infection, which is independent of the induction of many other host defense genes.

To determine whether such a role of FMO-2/FMO5 requires its catalytic activity, we used CRISPR-mediated genome editing to modify key conserved residues in the FMO-2/FMO5 FAD-binding domain, the NADPH-binding domain, or both (*Figure 8—figure supplement 2A–B*). Due to their conservation in FMOs from yeast, plants, and animals (*Figure 8—figure supplement 2B*), these residues are predicted to be required for electron transfer from organic substrates to cofactors FAD and NADPH (*Kubo et al., 1997*; *Rescigno and Perham, 1994*). Remarkably, mutations of either cofactor binding motif caused infection survival defects that were barely weaker than that of null *fmo-2/FMO5* mutants. Simultaneous mutation of both motifs produced susceptibility that was indistinguishable from that of *fmo-2/FMO5* null mutants (*Figure 8D*). These results suggested that both cofactor-binding sites were required for FMO-2/FMO5 function in host defense. In contrast, none of these mutations, alone or in combination, altered total lifespan on nonpathogenic *E. coli* (*Figure 8E*). Together, these data indicated that FMO-2/FMO5 catalytic activity may be specifically required for host defense against infection.

Simultaneous deletion of *nhr-49/PPARA* and *fmo-2/FMO5* resulted in similarly impaired infection survival as the single mutants (*Figure 9A*), suggesting that *nhr-49/PPARA* and *fmo-2/FMO5* function in the same pathway. On nonpathogenic *E. coli,* the lifespan defect of *nhr-49(-)* mutants was slightly improved by introduction of *fmo-2(-)* (*Figure 9B*). In contrast, introduction of *fmo-2(-)* in the two *nhr-49(gof)* mutant backgrounds almost completely suppressed the enhanced infection survival (*Figure 9C*), while their lifespan on nonpathogenic *E. coli* was also suppressed (for gf1) or slightly impaired (for gf2) (*Figure 9D*). These results show that that *fmo-2/FMO5* is absolutely required for the enhancement of host infection survival by *nhr-49/PPARA* gain-of-function, consistent with *fmo-2/FMO5* acting downstream of *nhr-49/PPARA*, while its effect on their lifespan is allele-specific.

In addition, we found that intestinal-restricted *fmo-2/FMO5* overexpression was sufficient to boost infection survival (*Figure 8—figure supplement 3A*). Interestingly, the lifespan of these animals was also extended on nonpathogenic *E. coli* (*Figure 8—figure supplement 3B*), confirming prior work (*Leiser et al., 2015*). These results suggested that elevating FMO-2/FMO5 levels in the intestine alone confers benefits not only in host defense but also against aging, possibly by increasing host resistance to food *E. coli* pathogenesis late in life (*McGee et al., 2011*; *Zhao et al., 2017*). Altogether, these observations suggested that *fmo-2/FMO5* is necessary and sufficient for host defense against *S. aureus*.

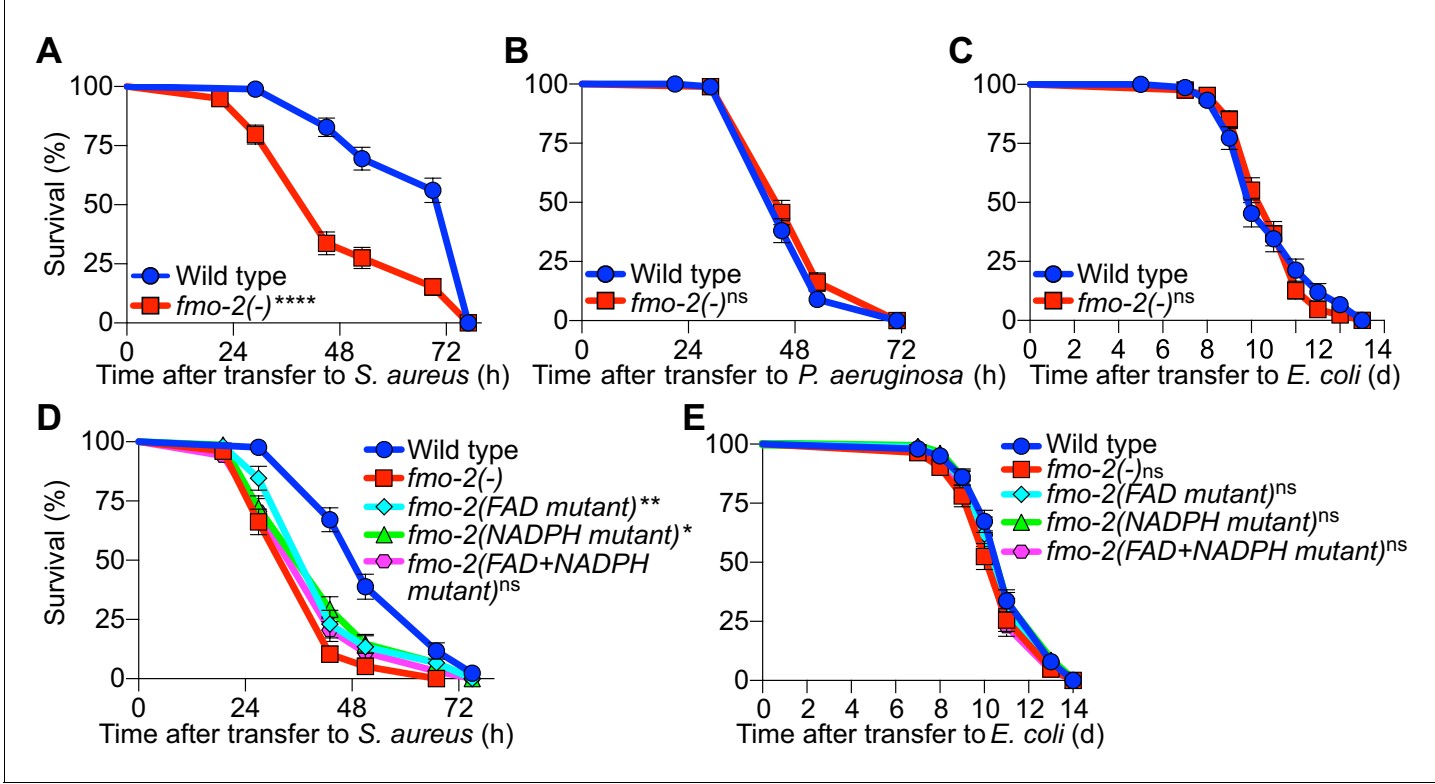

**Figure 8.** FMO-2/FMO5 is required for host survival of infection. (**A**) Survival of wild type and *fmo-2/FMO5* loss-of-function mutants infected with *S. aureus* SH1000. Data are representative of five independent trials. ****p≤0.0001 (Log-Rank test). (**B**) Survival of wild type and *fmo-2/FMO5* loss-of-function mutants infected with *P. aeruginosa* PA14. Data are representative of two independent trials. ns = not significant (Log-Rank test). (**C**) Lifespan of wild type and *fmo-2/FMO5* loss-of-function mutants fed nonpathogenic *E. coli* OP50. Data are representative of three independent trials. ns = not significant (Log-Rank test). (**D**) Survival of wild type, *fmo-2/FMO5*, *fmo-2(FAD)*, *fmo-2(NADPH)*, and *fmo-2(FAD+NADPH)* mutants infected with *S. aureus* SH1000. Data are representative of three independent trials. *p≤0.05, **p≤0.01, ns = not significant (Log-Rank test). Comparisons shown are between *fmo-2(-)* and the catalytic mutants. (**E**) Lifespan of wild type, *fmo-2/FMO5*, *fmo-2(FAD)*, *fmo-2(NADPH)*, and *fmo-2(FAD +NADPH)* mutants on *E. coli* OP50. Data are representative of 4 independent replicates. ns = not significant (Log-Rank test).

The online version of this article includes the following source data and figure supplement(s) for figure 8:

**Source data 1.** Survival of wild type and *fmo-2(-)* mutant animals infected with *S. aureus*.

**Source data 2.** Survival of wild type and *fmo-2(-)* mutant animals infected with *P. aeruginosa*.

**Source data 3.** Lifespan of wild type and *fmo-2(-)* mutant animals fed nonpathogenic *E. coli*.

**Source data 4.** Survival of wild type, *fmo-2(-)*, *fmo-2(FAD)*, *fmo-2(NADPH)*, and *fmo-2(FAD +NADPH)* mutant animals infected with *S. aureus*.

**Source data 5.** Lifespan of wild type, *fmo-2(-)*, *fmo-2(FAD)*, *fmo-2(NADPH)*, and *fmo-2(FAD +NADPH)* mutant animals on nonpathogenic *E. coli*.

**Figure supplement 1.** FMO-2/FMO5 is not required for the expression of a set of host defense genes.

**Figure supplement 1—source data 1.** *H02F09.3* mRNA levels in wild type and *fmo-2(-)* animals fed nonpathogenic *E. coli* or infected with *S. aureus*.

**Figure supplement 1—source data 2.** *ech-9* mRNA levels in wild type and *fmo-2(-)* animals fed nonpathogenic *E. coli* or infected with *S. aureus*.

**Figure supplement 1—source data 3.** *Y47H9C.1* mRNA levels in wild type and *fmo-2(-)* animals fed nonpathogenic *E. coli* or infected with *S. aureus*.

**Figure supplement 1—source data 4.** *C50F7.5* mRNA levels in wild type and *fmo-2(-)* animals fed nonpathogenic *E. coli* or infected with *S. aureus*.

**Figure supplement 1—source data 5.** *clec-52* mRNA levels in wild type and *fmo-2(-)* animals fed nonpathogenic *E. coli* or infected with *S. aureus*.

**Figure supplement 1—source data 6.** *Y65B4BR.1* mRNA levels in wild type and *fmo-2(-)* animals fed nonpathogenic *E. coli* or infected with *S. aureus*.

**Figure supplement 1—source data 7.** *hsp-17* mRNA levels in wild type and *fmo-2(-)* animals fed nonpathogenic *E. coli* or infected with *S. aureus*.

**Figure supplement 1—source data 8.** *srr-6* mRNA levels in wild type and *fmo-2(-)* animals fed nonpathogenic *E. coli* or infected with *S. aureus*.

**Figure supplement 1—source data 9.** *pals-39* mRNA levels in wild type and *fmo-2(-)* animals fed nonpathogenic *E. coli* or infected with *S. aureus*.

**Figure supplement 2.** Highly conserved amino acid sequences in FMO-2/FMO5.

**Figure supplement 3.** Intestinal overexpression of FMO-2/FMO5 boosts host survival of *S. aureus* infection.

**Figure supplement 3—source data 1.** Survival of wild type and intestinal overexpression (OE) line of *fmo-2* infected with *S. aureus*.

**Figure supplement 3—source data 2.** Lifespan of wild type and intestinal overexpression (OE) line of *fmo-2* on nonpathogenic *E. coli*.

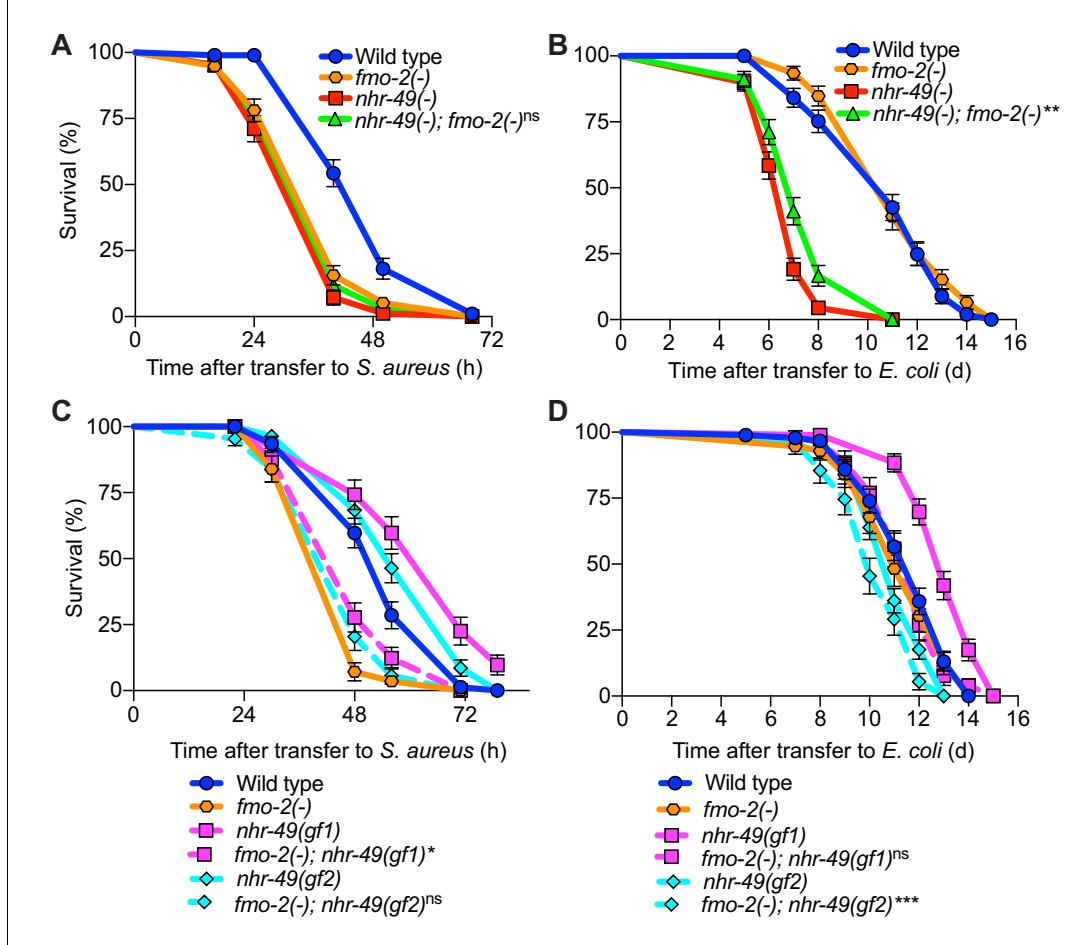

**Figure 9.** *fmo-2/FMO5* and *nhr-49/PPARA* function in the same genetic pathway. (**A**) Survival of wild type, *nhr-49(-)*, and *nhr-49(-);fmo-2(-)* animals infected with *S. aureus*. Data are representative of two independent trials. ns = not significant (Log-Rank test). (**B**) Lifespan of wild type, *nhr-49(-)*, and *nhr-49(-);fmo-2(-)* animals on nonpathogenic *E. coli*. Data are representative of two independent trials. \*\*p≤0.01 (Log-Rank test, compared to *nhr-49(-)*). (**C**) Survival of wild type, *fmo-2(-)*, *nhr-49(gf1)*, *fmo-2(-); nhr-49(gf1)*, *nhr-49(gf2)*, and *fmo-2(-); nhr-49(gf2)* animals infected with *S. aureus*. Data are representative of two independent trials. \*p≤0.05, ns = not significant (Log-Rank test, comparisons are between *fmo-2(-)* single and *fmo-2(-); nhr-49(gf)* double mutants). (**D**) Lifespan of wild type, *fmo-2(-)*, *nhr-49(gf1)*, *fmo-2(-); nhr-49(gf1)*, *nhr-49(gf2)*, and *fmo-2(-); nhr-49(gf2)* animals on nonpathogenic *E. coli*. Data are representative of two independent trials. \*\*\*p≤0.001, ns = not significant (Log-Rank test, comparisons are between *fmo-2(-)* single and *fmo-2(-); nhr-49(gf)* double mutants).

The online version of this article includes the following source data and figure supplement(s) for figure 9:

**Source data 1.** Survival of wild type, *nhr-49(-)*, and *nhr-49(-);fmo-2(-)* animals infected with *S. aureus*.

**Source data 2.** Lifespan of wild type, *nhr-49(-)*, and *nhr-49(-);fmo-2(-)* animals on nonpathogenic *E. coli*.

**Source data 3.** Survival of wild type, *fmo-2(-)*, *nhr-49(gf1)*, *fmo-2(-);nhr-49(gf1)*, *nhr-49(gf2)*, and *fmo-2(-); nhr-49(gf2)* animals infected with *S. aureus*.

**Source data 4.** Lifespan of wild type, *fmo-2(-)*, *nhr-49(gf1)*, *fmo-2(-);nhr-49(gf1)*, *nhr-49(gf2)*, and *fmo-2(-);nhr-49(gf2)* animals on nonpathogenic *E. coli*.

**Figure supplement 1.** *K08C7.4* is dispensable for host defense.

**Figure supplement 1—source data 1.** Survival of wild type and *nhr-49(-)* animals, infected with *S. aureus*, fed on either control (empty vector) RNAi, or RNAi against *K08C7.4*.

**Figure supplement 1—source data 2.** Survival of wild type, *nhr-49(gf1)*, and *nhr-49(gf2)* animals, infected with *S. aureus*, fed on either control (empty vector) RNAi, or RNAi against *K08C7.4*.

Since *K08C7.4/nfds-1* induction by infection was also dependent on NHR-49/PPAR-α (***Figure 3— figure supplements 2*** and ***3***), we examined genetic interactions between *nhr-49/PPARA* and *K08C7.4/nfds-1* for *S. aureus* infection survival. *K08C7.4/nfds-1* RNAi did not affect the phenotypes of wild type or *nhr-49/PPARA* null mutants compared with control RNAi animals (***Figure 9—figure supplement 1A***). Similarly, *K08C7.4/nfds-1* RNAi had no effect on *nhr-49(gf2)* animals, and only a mild suppressive effect in *nhr-49(gf1)* animals (***Figure 9—figure supplement 1B***). Altogether, these

observations suggest that *K08C7.4/nfds-1* may be dispensable for host defense and may play a minor role downstream of *nhr-49/PPARA*.

## Discussion

Because bacteria serve as nutritional source for *C. elegans,* and because intestinal infections cause destruction of the epithelium resulting in loss of nutrient absorption, transcriptional responses to nutritional challenges are likely intertwined with the transcriptional host defense response to infection itself. This raises the question of whether *C. elegans* senses infection as a stress per se, through its physiological consequences in the organism, or a combination of both. Here, by directly comparing transcriptomes of animals that were infected with *S. aureus* or were starved, we discovered that starvation and infection elicit large and distinct transcriptional signatures. This indicates that the *C. elegans* host response to *S. aureus* infection is not entirely the result of starvation, and enables the dissection of infection-specific and starvation-specific host response regulatory modules as shown here.

In the present study, we found that loss of HLH-30/TFEB almost completely abrogated differential gene expression between starvation and infection – implicating HLH-30/TFEB not just in a hypothetical overlapping response but also in each of these two distinct signatures. This strongly suggests that HLH-30/TFEB integrates metabolic and other stresses to contribute to stress-specific transcriptional responses. The molecular mechanisms that enable a single transcription factor to mediate specific transcriptional responses to distinct stresses may involve stress-specific signals or transcriptional co-factors.

By focusing on *fmo-2/FMO5,* which we first showed was highly and specifically induced by *S. aureus* infection (*Irazoqui et al., 2008*; *Irazoqui et al., 2010a*) and is only partially dependent on HLH-30/TFEB (*Visvikis et al., 2014*), we discovered a novel role for the nuclear receptor NHR-49/PPAR-α in host defense against *S. aureus* infection. This discovery is related to prior studies that showed that *Enterococcus faecalis,* an enteric Gram-positive human pathogen unrelated to *S. aureus*, induces *fmo-2/FMO5* during infection of *C. elegans* (*Dasgupta et al., 2020*). In the same work, RNAi of *nhr-49/PPARA* resulted in enhanced susceptibility to *P. aeruginosa, S. enterica* serovar Typhimurium, and *Candida albicans,* but not *S. aureus* (*Dasgupta et al., 2020*). These results suggest that *nhr-49/PPARA* may play other roles in host defense against infection, aside from the induction of *fmo-2/FMO5* (which is not induced by *P. aeruginosa* [*Figure 3—figure supplement 1* and *Irazoqui et al., 2008*, *Irazoqui et al., 2010a*]). Moreover, *nhr-49/PPARA* RNAi did not abrogate P*fmo-2::gfp* expression in the pharynx (*Dasgupta et al., 2020*). The reasons of these discrepancies with our observations remain unclear, but could be related to the use of RNAi instead of null alleles or to pathogen-intrinsic differences between *S. aureus* and *E. faecalis*. Nonetheless, *fmo-2/FMO5* was first described as the most highly induced gene in *E. faecalis* -infected animals compared with *E. coli* controls several years ago (*Wong et al., 2007*; *Yuen and Ausubel, 2018*). Subsequent work showed that *nhr-49/PPARA* silencing impaired infection survival of *E. faecalis* (*Sim and Hibberd, 2016*), but the connection between *nhr-49/PPARA* and *fmo-2/FMO5* during infection was not established until our present work and that of others (*Dasgupta et al., 2020*). Thus, NHR-49/PPAR-α appears to play an important role in defense against a broad range of bacteria, a conclusion that is reinforced by the recent identification of small molecules that protect germline-defective *C. elegans* from *P. aeruginosa* -mediated killing via NHR-49/PPAR-α (*Hummell et al., 2021*).

Recently, NHR-49/PPAR-α was shown to mediate the defense response to exogenous oxidative stress (*Goh et al., 2018*; *Hu et al., 2018*). Thus, NHR-49/PPAR-α participates in host defense against biotic and abiotic stressors, and should be considered a key player in the organismal stress response alongside SKN-1/NRF2, DAF-16/FOXO3, and HLH-30/TFEB (*Blackwell et al., 2015*; *Lin et al., 2018*; *Tissenbaum, 2018*). Our analysis showed that NHR-49/PPAR-α is not required for as large a portion of the host response to infection as HLH-30/TFEB, even though NHR-49/PPAR-α is partially required for HLH-30/TFEB induction. The larger HLH-30/TFEB regulon implies that during infection signals in addition to NHR-49/PPAR-α activation contribute to HLH-30/TFEB regulation. Similar to HLH-30/TFEB, how NHR-49/PPAR-α induces specific responses to distinct stresses is also unknown. These findings are relevant beyond nematodes, as PPAR-α regulates TFEB in mammalian cells (*Kim et al., 2017*). Moreover, the microbiota represses HNF4-α, a second NHR-49 homolog, in zebrafish and mice, to maintain intestinal homeostasis (*Davison et al., 2017*). Therefore, unraveling the control of

NHR-49/PPAR-α in relation to intestinal microbiota and infection may provide useful information to understand vertebrate intestinal homeostasis and host defense.

NHR-49/PPAR-α is a known regulator of fat metabolism during starvation (*Van Gilst et al., 2005*). Lipid metabolism plays an important role in immune activation and host defense in *C. elegans* (*Anderson et al., 2019*; *Nandakumar and Tan, 2008*). Recent work showed that infection by *E. faecalis* results in NHR-49/PPAR-α-dependent upregulation of lipid catabolism and downregulation of lipid synthesis genes (*Dasgupta et al., 2020*). Whether this is also true of *S. aureus* infection is unknown, but our RNA-seq results suggest an enrichment of genes related to 'Lipid Metabolic Process' – including catabolism and anabolism, in infected wild type but not *nhr-49/PPARA* mutants. Thus, a mechanism of NHR-49/PPAR-α-mediated host defense could be by the regulation of fat metabolism, in parallel to induction of *fmo-2/FMO5*.

In addition to leading us to discover NHR-49/PPAR-α, FMO-2/FMO5 is interesting in its own right. Regulation of *fmo-2/FMO5* is complex. During infection, NHR-49/PPAR-α appeared to be essential for *fmo-2/FMO5* expression in the whole animal, while *hlh-30/TFEB* was partially dispensable in the intestine and pharynx. Hypoxia and dietary restriction induce *fmo-2/FMO5* (*Leiser et al., 2015*; *Shen et al., 2005*), and so do gain-of-function mutations in *hif-1/HIF1* or *skn-1/NRF2* (*Leiser et al., 2015*; *Nhan et al., 2019*). Lifespan extension by dietary restriction or *hif-1/HIF1* gain of function requires *fmo-2/FMO5*; moreover, *hlh-30/TFEB* loss of function quenches lifespan extension by *hif-1/HIF1* and *fmo-2/FMO5* induction by hypoxia and fasting (*Leiser et al., 2015*). These observations lend further support to our findings that HLH-30/TFEB partially induces *fmo-2/FMO5* during infection.

In addition, we found that loss of FMO-2/FMO5 causes a severe defect in infection survival without affecting longevity. Thus, FMO-2/FMO5 represents a novel host defense effector. We previously examined the requirement for *fmo-2/FMO5* using RNAi-mediated silencing, but such manipulation failed to produce a phenotype for reasons unknown (*Irazoqui et al., 2010a*). Moreover, the failure of tissue-specific RNAi to elicit a phenotype and the toxicity of *fmo-2/FMO5* extrachromosomal transgenic constructs precluded our investigation of the tissues of *fmo-2/FMO5* action for host defense. However, single copy intestinal expression of FMO-2/FMO5 boosted host defense, suggesting that FMO-2/FMO5 could play a major role in the intestine, a hub for host defense in *C. elegans* (*McGhee, 2007*). Nonetheless, FMO-2/FMO5 induction appears to be a major mechanism of host defense in *C. elegans*. Exactly how FMO-2/FMO5 promotes host infection survival is poorly understood, but site-directed mutagenesis of the NADPH and FAD-binding sites revealed that the mechanism of action requires its catalytic activity. In addition, human FMO5 can generate large amounts of $H_2O_2$ from $O_2$ (*Fiorentini et al., 2016*). Thus, it is possible that FMO-2/FMO5 is an infection-specific NADPH oxidase that generates $H_2O_2$ with antimicrobial and signaling functions (*McCallum and Garsin, 2016*; *Sies and Jones, 2020*). The observed roles of *fmo-2/FMO5* in survival of heat, di-thiothreitol, and tunicamycin stresses are consistent with a $H_2O_2$-mediated signaling role (*Leiser et al., 2015*). Alternatively, ER localization of FMOs may be important for regulation of ER stress or of the UPR[ER]. Several infections induce ER stress, and the UPR[ER] promotes host defense in many organisms (*Choi and Song, 2019*). In support of this, yeast FMO (yFMO), which localizes to the ER membrane, is activated by the UPR[ER] and is required for protein folding in the ER (*Suh and Robertus, 2000*; *Suh et al., 1999*). Moreover, the five human FMOs localize to the ER membranes of cells of the liver, lung, and kidney (*Dolphin et al., 1996*; *Phillips et al., 1995*). Human FMO5 is also expressed in the intestine (*Hernandez et al., 2004*; *Zhang and Cashman, 2006*) and murine FMO5 is expressed in the epithelial cells of small and large intestine (*Scott et al., 2017*). While its subcellular localization in intestinal epithelial cells was not characterized, it is possible that FMO5 localizes to their ER membrane (*Phillips et al., 1995*). Further research is necessary to understand the precise connections linking infection, ER stress, and FMO function in nematodes and higher organisms.

Despite the paucity of information, FMOs are emerging as important host defense factors across phylogeny. In plants, FMO1 is required for the conversion of pipecolic acid to N-hydroxypipecolic acid, which provides systemic acquired resistance to bacterial and oomycete infections (*Hartmann et al., 2018*). In mammals, FMO3 is an evolutionarily ancient FMO that exhibits unique substrate specificity and catalyzes multiple drugs that is important for their detoxification (*Krueger and Williams, 2005*). Moreover, mammalian FMO proteins were recently shown to promote cellular stress resistance and alter cellular metabolism. Overexpression of each of the five

mouse FMOs (FMO1-FMO5) in human hepatocytes and kidney epithelial cells conferred resistance to $Cd^{2+}$, arsenite, paraquat, UV radiation, and rotenone (*Huang et al., 2021*). Additionally, FMO overexpression increased mitochondrial respiration with concomitant decrease in glycolytic activity (*Huang et al., 2021*). However, to date no reports have indicated an important role for FMO5, or other FMOs, in mammalian (or any animal) innate immunity. In mice, FMO5 is expressed in many tissues and organs, including the liver and the epithelium of the gastrointestinal tract (*Scott et al., 2017*). Mouse FMO5 is required for sensing the microbiota, and $Fmo5^{-/-}$ mutants exhibit altered metabolic profiles and microbiomes compared with wild-type mice (*Scott et al., 2017*). Furthermore, $Fmo5^{-/-}$ mutants exhibit a 70% reduction in plasma TNF-α compared with wild type (*Scott et al., 2017*). Together, these observations suggest that FMO5 is an important microbiota sensor and effector that modulates the intestinal microbiota, but the mechanism of action is unknown. Therefore, elucidation of mechanisms of host defense mediated by FMO-2 in nematodes and FMO5 in mammals will provide fundamental insight into evolutionarily conserved mechanisms of host defense against infection and identify therapeutic opportunities for infections and inflammatory diseases.

## Materials and methods

### Experimental model

The nematode *C. elegans* was used as the experimental model for this study. Strains were maintained at 15–20°C on Nematode Growth Media (NGM) plates seeded with $Str^R$ *E. coli* OP50-1 strain using standard methods (*Stiernagle, 2006*).

### Method details

Infection assays

*S. aureus* SH1000 strain was grown overnight in tryptic soy broth (TSB) containing 50 µg/ml kanamycin (KAN). Overnight cultures were diluted 1:1 with TSB and 10 µl of the diluted culture was uniformly spread on the entire surface of 35 mm tryptic soy agar (TSA) plates containing 10 µg/ml KAN. Plates were incubated for 5–6 hr at 37°C, then stored overnight at 4°C. *P. aeruginosa* isolate PA14 was grown overnight in Luria broth. 10 µl of the overnight culture was uniformly spread on the entire surface of 35 mm NGM plates. Plates were incubated at 37°C for 24 hr followed by 25°C for 48 hr (*Powell and Ausubel, 2008*). Animals were treated with 100 µg/ml 5-fluoro-2'-deoxyuridine (FUDR) at L4 larval stage for ~24 hr at 15–20°C before transfer to *S. aureus* or *P. aeruginosa* plates. Three plates were assayed for each strain in each replicate, with 20–40 animals per plate. Infection assays were carried out at 25°C except for the RNAi experiments. We found that feeding wild type animals with *E. coli* strain HT115(DE3) prior to putting them on SH1000 made them die much faster than when fed on regular *E. coli* OP50 food. Therefore, to determine any differences in survival to SH1000 upon RNAi of the target genes, infection assays were carried out at 20°C to slow down the growth of *S. aureus*. Survival was quantified using standard methods (*Powell and Ausubel, 2008*). Animals that crawled off the plate or died of bursting vulva were censored. Infection assays were carried out at least twice.

### *S. aureus* infection for RNA analysis

To prepare infection plates, *S. aureus* SH1000 was grown overnight in TSB containing 50 µg/ml KAN. 500–1000 µl of overnight culture was uniformly spread on the entire surface of freshly prepared 100 mm TSA plates supplemented with 10 µg/ml KAN. The plates were incubated for 6 hr at 37°C, then stored overnight at 4°C. To prepare *P. aeruginosa* plates, *P. aeruginosa* isolate PA14 was grown overnight in Luria broth. 1 ml of overnight culture was uniformly spread on the entire surface of freshly prepared 100 mm NGM plates. The plates were first incubated at 37°C for 24 hr and then at 25°C for 48 hr. To prepare control plates with nonpathogenic *E. coli*, 1 ml of 10–20X concentrated overnight culture of OP50-1 bacteria was spread on 100 mm NGM plates, incubated for 5–6 hr at 37°C, and then stored at 4°C, similar to *S. aureus* plates. To prepare plates for starvation, TSA plates were treated similarly to infection plates, except that nothing was added to them. Synchronized young adults of wild type and mutants were seeded the next day on *S. aureus*, *P. aeruginosa*, OP50-1, and starvation plates that were previously warmed to room temperature. After 4 hr incubation at

25°C, animals for all conditions were washed 3–4 times in water, and then lysed in 1 ml of TRIzol reagent (Invitrogen). The samples were snap frozen in liquid nitrogen, then stored at −80°C. RNA was extracted using 1–bromo–3–chloropropane (MRC) followed by purification with isopropanol-ethanol precipitation. RNA was analyzed by qPCR or sequencing. In some experiments, the magnitude of *fmo-2* expression measured by RT-qPCR showed some inter-trial variation. The reasons for this are not fully understood, but could be related to the very low expression in infected animals or to variation in environmental variables, such as plate batch. Regardless, the results strongly support our conclusions regarding *hlh-30* and *nhr-49* dependency. For sequencing, RNA was additionally purified using PureLink RNA Mini Kit (Invitrogen). Four independent biological replicates were submitted to BGI for library preparation and sequencing using BGI-seq 500.

## Quantitative RT-PCR

After each treatment, *C. elegans* were collected in sterile water and lysed using TRIzol Reagent (Invitrogen). Total RNA was extracted and purified as described before and then digested with DNAse (Bio-Rad). 100–1000 ng of total RNA was used for cDNA synthesis using iScript cDNA synthesis kit (Bio-Rad). RT-qPCR was performed using SYBR Green Supermix (Bio-Rad) using a ViiA7 Real-Time qPCR system (Applied Biosystems). Primer sequences are provided in *Supplementary file 5*. At least two independent biological replicates were used for each treatment and *C. elegans* strain. qPCR Ct values were normalized to the *snb-1* control gene, which did not change with the conditions tested, to calculate RT-qPCR ΔCt values. Data analysis was carried out using the *Pfaffl, 2001* method. Heat maps were generated using open access online tool Morpheus (https://software.broadinstitute.org/morpheus).

## RNA sequencing analysis

BGI provided clean reads in FASTQ format. Clean FASTQ files were verified using FastQC (https://www.bioinformatics.babraham.ac.uk/projects/fastqc) using Bioconductor in RStudio (*Loraine et al., 2015*) and used as input for read mapping in Salmon v.0.9.1 (*Patro et al., 2017*) using WBCel.235.cdna from Ensembl (http://www.ensembl.org) as reference transcriptome. Salmon outputs in quant format were used for input in DESeq2 (*Love et al., 2014*) in Bioconductor in RStudio for count per gene estimation using batch correction. Total counts per gene tables from DESeq2 were used as input for DEBrowser (*Kucukural et al., 2019*) for verification of transcriptome replicate similarity, data analysis using the built-in DESeq2 algorithm for differential gene expression analysis (adjusted p value ≤ 0.01 was considered significant), visualization, and interactive data mining. Overlap between gene sets was determined using the Venn tool in BioInfoRx (https://bioinforx.com). GO representation analysis was performed using online tool g:Profiler (*Raudvere et al., 2019*) (https://biit.cs.ut.ee/gprofiler/gost).

## Longevity (aging) assays

Animals were transferred to 60 mm NGM plates seeded with 10–20X concentrated *E. coli* OP50-1 bacteria supplemented with 100 µg/ml FUDR. For consistency with infection assays, longevity assays were also performed at 25°C. Three plates were assayed for each strain in each replicate, with 20–40 animals per plate. Experiments were repeated at least twice. Animals that did not respond to prodding were scored as dead, and the animals that died from bursting vulva or crawled off the plate were censored.

## RNA interference (RNAi)

Knockdown of genes was carried out by feeding *C. elegans* the *E. coli* strain HT115(DE3) expressing double-stranded RNA against the target gene using standard methods (*Kamath et al., 2001*). Briefly, glycerol stocks of RNAi clones containing the appropriate vectors were streaked out on LB media plates containing ampicillin (50 µg/ml) and tetracycline (10 µg/ml) and the bacteria were allowed to grow at 37°C for 16 hr. Single colonies of the clones were then grown in LB liquid media containing ampicillin (50 µg/ml) for 16 hr at 37°C with constant shaking. The cultures were concentrated 10X and 300–500 µl were plated on NGM plates containing ampicillin (50 µg/ml) and IPTG (1 mM). Plates were dried in a fume hood and left overnight at room temperature for dsRNA induction. Next day, five L4 animals were placed on the plates and then allowed to grow at 20°C for 4 days to

get enough L4 progeny of the next generation. FUDR (100 µg/ml) was placed on the plates and next day young adult animals were picked and placed on TSA plates containing uniformly spread SH1000. Survival on SH1000 was performed as described above in Infection Assays. For assessing lifespan after RNAi of the target genes, L4 animals were picked and placed on OP50 containing NGM plates supplemented with 100 µg/ml FUDR and then placed at 25°C. Lifespan was carried out as described above in Longevity (aging) assays.

## Generation of transgenic strains

To construct P*nhr-49::nhr-49::gfp* containing plasmid, a 6.6 kb genomic fragment of *nhr-49* gene (comprising of 4.4 kb coding region covering all *nhr-49* transcripts plus 2.2 kb sequence upstream of ATG) was cloned into the GFP expression vector pPD95.77 (Addgene #1495), as reported previously (*Ratnappan et al., 2014*). For generating tissue-specific constructs, the *nhr-49* promoter was replaced with tissue-specific promoters using *Sbf*I and *Sal*I restriction enzymes. The primers that were used to amplify tissue-specific promoters are listed in *Supplementary file 5*. For the generation of rescue strains, each rescue plasmid (100 ng/µl) was injected along with pharyngeal muscle-specific P*myo-2::mCherry* co-injection marker (25 ng/µl) into *nhr-49(nr2041)* mutant strain, using standard methods (*Mello and Fire, 1995*). Strains were maintained by picking animals that were positive for both GFP and mCherry.

To construct P*K08C7.4::gfp* containing plasmid, a 2 kb 5' upstream sequence (upstream of ATG) was amplified from *C. elegans* genomic DNA, and then cloned into the GFP expression vector pPD95.75 (Addgene #1494) using Gibson Assembly (*Gibson et al., 2009*). Promoter and vector sequences were assembled using Gibson Assembly Kit (NEB #E2611). For the generation of transgenic strains, P*K08C7.4::gfp* containing plasmid was injected into wild-type animals (100 ng/µl) along with *rol-6(su1006)* con-injection marker (for a total of 700 ng), using standard methods (*Mello and Fire, 1995*). Strains were maintained by picking animals that were rollers as well as positive for GFP. The primers that were used to amplify *K08C7.4* promoter and pPD95.75 vector sequences used in the Gibson Assembly are listed in *Supplementary file 5*.

*fmo-2(FAD)*, *fmo-2(NADPH)*, and *fmo-2(FAD+NADPH)* strains were generated using CRISPR-Cas9 genome editing as described (*Dokshin et al., 2018*). Residues for mutation were selected based on protein sequence alignment and as previously reported (*Bartsch et al., 2006*). To isolate worms with mutated residue(s) in FAD or NADPH motifs, silent mutations that resulted in restriction enzyme sites (*Pvu*II for FAD, and *Ava*II for NADPH) without any change in the amino acid(s) were created in the repair templates. A PCR fragment spanning the mutated nucleotides was amplified from the progeny of the injected worms, followed by digestion with the above-mentioned restriction enzymes. Mutations in FAD and NADPH motifs were confirmed by sequencing PCR fragments amplified from the corresponding regions in the mutant animals. To generate *fmo-2(FAD+NADPH)* double mutant, the *fmo-2(NADPH)* mutant strain was used as a background for a second round of CRISPR microinjections. Sequences for crRNAs, repair templates, and the genotyping primers used for the construction of these strains are listed in *Supplementary file 5*.

## Image analysis

Images were captured using a Lionheart FX Automatic Microscope (BioTek Instruments) under a 4X or 20X objective. Twenty to 50 animals were anesthetized using 20–100 mM NaN$_3$ on a 2% agarose pad immediately prior to imaging. Comparable images were captured with the same exposure time and magnification. Fluorescence microscopy analysis was independently replicated at least three times. Fluorescence intensity was quantified using ImageJ. Intestinal fluorescent intensity was quantified by outlining the intestine in the DIC images and then quantifying the corresponding fluorescent signals in the GFP images using ImageJ. There are several potential factors that may cause differences between RT-qPCR of *fmo-2* transcript and quantification of images of the P*fmo-2::gfp* transcriptional reporter: (1) fluorescence microscopy does not possess the same sensitivity and linear range as RT-qPCR, which is the preferred method for expression analysis; (2) the P*fmo-2::gfp* transcriptional reporter is a transgene that only contains 1536 bp of upstream (promoter) sequence plus the first 16 codons of *fmo-2* (*Goh et al., 2018*), and thus could be missing regulatory elements that are present in the endogenous (chromosomal) locus; (3) steady-state mRNA levels (measured by RT-qPCR) are the net result of the rates of transcription and degradation, and so mRNA stabilization

could play a role in increased *fmo-2* expression, which would not be reproduced by the GFP reporter.

## Quantification and statistical analysis

Prism 8 (GraphPad) was used for statistical analyses. Survival data were compared using the Log-Rank (Mantel-Cox) test. A p value $\leq 0.05$ was considered significantly different from control. For comparisons to a single reference, two-sample, two-tailed *t* tests were performed to evaluate differences between $\Delta$Ct values (*Schmittgen and Livak, 2008*). For multiple comparisons, statistical significance was examined by one-way ANOVA followed by Šídák's post-hoc test. A p value $\leq 0.05$ was considered significant.

## Acknowledgements

The authors are grateful to members of the Irazoqui laboratory, the Program in Innate Immunity, and the Department of Microbiology and Physiological Systems for helpful insights and discussions. Joyce Barrett, Linda Benson, Richard Fish, Amy Parker, Cheryl Barry, and Tammy Bailey provided expert facilities and administrative assistance. Scott F Leiser (University of Michigan) provided KAE11 strain. Some strains used in this study were provided by the *Caenorhabditis* Genetics Center, which is funded by the NIH Office of Research Infrastructure Programs (P40-OD010440). Research reported in this publication was supported by the National Institute of General Medical Sciences of the National Institutes of Health under award number GM101056, and by the National Science Foundation under award number NSF1457055 (to JEI), and by a grant from the National Institutes of Aging (R01AG051659) to AG. The content is solely the responsibility of the authors and does not necessarily represent the official views of the National Institutes of Health.

## Additional information

### Funding

| Funder | Grant reference number | Author |
| --- | --- | --- |
| National Institute of General Medical Sciences | R01-GM101056 | Javier E Irazoqui |
| National Science Foundation | NSF1457055 | Javier E Irazoqui |
| NIA | R01AG051659 | Arjumand Ghazi |
| Canadian Institutes of Health Research (CIHR) | PJT-153199 | Stefan Taubert |

The funders had no role in study design, data collection and interpretation, or the decision to submit the work for publication.

### Author contributions

Khursheed A Wani, Conceptualization, Data curation, Formal analysis, Validation, Investigation, Visualization, Methodology, Writing - original draft, Writing - review and editing; Debanjan Goswamy, Validation, Investigation; Stefan Taubert, Arjumand Ghazi, Resources, Writing - review and editing; Ramesh Ratnappan, Resources; Javier E Irazoqui, Conceptualization, Resources, Data curation, Formal analysis, Supervision, Funding acquisition, Visualization, Methodology, Writing - original draft, Project administration, Writing - review and editing

### Author ORCIDs

Khursheed A Wani (iD) https://orcid.org/0000-0003-3559-7962
Stefan Taubert (iD) https://orcid.org/0000-0002-2432-7257
Ramesh Ratnappan (iD) http://orcid.org/0000-0001-7055-9043
Arjumand Ghazi (iD) http://orcid.org/0000-0002-5859-4206
Javier E Irazoqui (iD) https://orcid.org/0000-0001-6553-1329

Decision letter and Author response
Decision letter https://doi.org/10.7554/eLife.62775.sa1
Author response https://doi.org/10.7554/eLife.62775.sa2

## Additional files

### Supplementary files

• Supplementary file 1. Differential gene expression analysis by RNA-seq comparing starvation with infection in wild-type animals (related to *Figure 1*).

• Supplementary file 2. Biological pathway over-representation analysis comparing starvation with infection in wild-type animals (related to *Figure 1*).

• Supplementary file 3. Differential gene expression analysis by RNA-seq comparing starvation with infection in *hlh-30(-)* animals (related to *Figure 2*).

• Supplementary file 4. Differential gene expression analysis by RNA-seq comparing starvation with infection in *nhr-49(-)* animals (related to *Figure 6*).

• Supplementary file 5. List of oligos, crRNAs, and repair templates used in this work.

• Transparent reporting form

### Data availability

RNA-seq reads are deposited in SRA (NCBI/NIH).

The following datasets were generated:

| Author(s) | Year | Dataset title | Dataset URL | Database and Identifier |
|---|---|---|---|---|
| Wani KA, Irazoqui JE | 2021 | *S. aureus* SH1000 vs Starvation transcriptional profiling (by RNA sequencing) in *C. elegans* and its dependence on HLH-30 | https://www.ncbi.nlm.nih.gov/bioproject/PRJNA727590 | NCBI BioProject, PRJNA727590 |
| Wani KA, Irazoqui JE | 2021 | *S. aureus* SH1000 vs Starvation transcriptional profiling (by RNA sequencing) in *C. elegans* and its dependence on NHR-49 | https://www.ncbi.nlm.nih.gov/bioproject/PRJNA728096 | NCBI BioProject, PRJNA728096 |

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
