## [Decision Letter]

**Acceptance summary:**

The direct comparison between dietary stress and pathological stress in this manuscript is extremely important, as the two are generally inseparable. These data, identifying differing roles for two key transcription factors and a key gene downstream of immune response, show that organisms have substantial overlapping and distinct responses to concurrent stresses. The results will form the basis for future work to further understand the common and separable mechanisms of related stress responses including but not limited to pathogenic stress.

**Decision letter after peer review:**

Thank you for submitting your article "NHR-49/PPAR-α and HLH-30/TFEB cooperate for *C. elegans* host defense via a flavin-containing monooxygenase" for consideration by *eLife*. Your article has been reviewed by 3 peer reviewers, and the evaluation has been overseen by a Reviewing Editor and Matt Kaeberlein as the Senior Editor. The following individual involved in review of your submission has agreed to reveal their identity: Katja Dierking (Reviewer #2).

The reviewers have discussed the reviews with one another and the Reviewing Editor has drafted this decision to help you prepare a revised submission.

As the editors have judged that your manuscript is of interest, but as described below that additional experiments and/or explanations are required before it is published, we would like to draw your attention to changes in our revision policy that we have made in response to COVID-19 (https://elifesciences.org/articles/57162). First, because many researchers have temporarily lost access to the labs, we will give authors as much time as they need to submit revised manuscripts. We are also offering, if you choose, to post the manuscript to bioRxiv (if it is not already there) along with this decision letter and a formal designation that the manuscript is "in revision at *eLife*". Please let us know if you would like to pursue this option. (If your work is more suitable for medRxiv, you will need to post the preprint yourself, as the mechanisms for us to do so are still in development.)

Summary:

In this manuscript, "NHR-49/PPAR-a and HLH-30/TFEB cooperate for *C. elegans* host defense via a flavin-containing monooxygenase" the authors study the infection-specific *C. elegans* response to Staphylococcus aureus. In particular, they present three important findings. First, they directly compare the pathogen response to the starvation response in the worm for the first time, producing a subsets of starvation-only and infection-only genes. This is relevant, and probably should have been done by the field years ago to better define the "pathogen response", because infected *C. elegans* are likely experiencing nutritional stress during infection because of the destruction of the intestinal tissue. Second, they show that HLH-30/TFEB is critical for host response to BOTH infection and starvation. They observed a startling lack of induced gene expression to both conditions in a hlh-30 mutant animal. However, a few genes were still induced, including fmo-2. The authors further investigate this HLH-30 independent regulation and identify the nuclear hormone receptor NHR-49 as its source. Tissue-specific expression analysis showed NHR-49 functions in multiple tissues to affect pathogen resistance. Further study of the FMO-2 effector protein, regulated by NHR-49, showed that it is necessary and sufficient to modify pathogen sensitivity, representing the third notable finding. Flavin-containing monooxygenases like FMO-2 have been reported in plants to be involved in pathogen defense, but this, along with another recent report, establishes this role in an animal.

The reviewers were in general agreement about the value of this work, especially the establishment of starvation vs. pathogenic responses, although there were some questions as to the choice of timepoint. The biggest concern, brought up in every review, was that the manuscript makes a number of claims about the novelty of the NHR-49 and FMO-2 findings, when NHR-49 was previously found to be a key player downstream of pathogenic stress in a 2016 paper, whereas FMO-2 was recently found to play a role in pathogenic stress protection in a paper published this summer. While these previous findings weaken the novelty of this manuscript and must be cited, there are other novel aspects remaining including the establishment of starvation vs. pathogenic responses, HLH-30/TFEB data and analysis, and additional tissue-specific details about NHR-49 and FMO-2, to warrant a revision. It is likely that by citing and downplaying the novelty of those findings that reinforce previous work, while highlighting the novel aspects of this work, in addition to responding effectively to each of the additional concerns of the reviewers, that this paper will be acceptable for publication in *eLife*. I will note that it is possible that most of the reviewer concerns can be addressed through the text, some may require additional experiments to adequately address, and thus I have selected the full revision BioRxiv option now used at *eLife*.

Essential revisions:

1. A primary criticism noticed by every reviewer is that parts of these results (nhr-49 and fmo-2 induction) have been published in a similar pathogenic environment, and that they did not cite these publications and made statements of priority that were inaccurate. This diminished novelty, although there are still other novel aspects remaining and if those aspects were highlighted and these aspects cited and properly characterized, the manuscript would be much improved.

2. The authors should discuss the processes (GO terms) found in their immune vs nutritional response data, and what the two responses share in common, in addition to the nhr-49 mutant data, as this would further increase the insightfulness and impact of this study.

3. Similarly, the authors find that NHR-49 is required for fmo-2 induction, but as NHR-49 is a known regulator of fat metabolism in response to starvation, this was not discussed in the context of their findings or the nhr-49 transcriptome data.

4. There is no explanation for the use of a 4h time period for post-infection and responses. Since this is a relatively short time, a justification or evidence that this time is appropriate is relevant, as previous studies have looked at longitudinal changes in starvation response, for example.

5. In the comprehensive analysis of genes induced by S. aureus infection, only 6 infection-responsive genes were HLH-30-independent and only 2 were NHR-49-dependent. This is very strange, especially since the two genes may share one promoter site (K08C7.4 and K08C7.5), and is not discussed or addressed in the text. How and why would a transcription factor bind only one site in response to a major stress? Similarly is there an effect of K08C7.4 knockdown / knockout? Thus the protective effect of nhr-49(gf) mutants should be diminished / abolished in fmo-2 KO background (and/or K08C7.4).

6. The author's data suggest that fmo-2 expression is not particularly induced in the hlh-30 mutant during S. aureus infection. (Compare Figure 3C and Figure 3D). In contrast, the qRT-PCR data demonstrate that the gene expression levels at this same time have risen >1000 fold in Figure S1 (Note, that in Figure 2D fmo-2 is only ~50-60-fold up. Please, also address this discrepancy). i.e. according to qRT-PCR, fmo-2 expression in the hlh-30 mutant on S. aureus should be similar to infected WT, while according to imaging it's much closer to nhr-49 mutant. This discrepancy between reporter and qRT-PCR needs to be discussed or rectified.

7. Why did authors exclude intestine from the list of tissues where nhr-49-mediated expression of fmo-2 occurs? High magnification images of the FMO-2 reporter strain in hlh-30 and nhr-49 mutants, where intestine-specific fluorescence can be quantified, would be helpful.

---

## [Author Response]

Essential revisions:1. A primary criticism noticed by every reviewer is that parts of these results (nhr-49 and fmo-2 induction) have been published in a similar pathogenic environment, and that they did not cite these publications and made statements of priority that were inaccurate. This diminished novelty, although there are still other novel aspects remaining and if those aspects were highlighted and these aspects cited and properly characterized, the manuscript would be much improved.

We completely agree with the Reviewers and apologize for the oversight. This has now been corrected. Among other references in the text, we now include a segment that specifically addresses the similarities and contrasts between our work presented here and that of Dasgupta et al. in the Discussion section (L 481–506):

“By focusing on *fmo-2/FMO5*, which we first showed was highly and specifically induced by *S. aureus* infection (Irazoqui et al., 2008, 2010a) and is only partially dependent on HLH-30/TFEB (Visvikis et al., 2014), we discovered a novel role for the nuclear receptor NHR-49/PPAR-α in host defense against *S. aureus* infection. This discovery is related to prior studies that showed that *Enterococcus faecalis*, an enteric Gram-positive human pathogen unrelated to *S. aureus*, induces *fmo-2/FMO5* during infection of *C. elegans* (Dasgupta et al., 2020). In the same work, RNAi of *nhr-49/PPARA* resulted in enhanced susceptibility to *P. aeruginosa*, *S. enterica* serovar Typhimurium, and *Candida albicans*, but not *S. aureus* (Dasgupta et al., 2020). These results suggest that *nhr-49/PPARA* may play other roles in host defense against infection, aside from the induction of *fmo-2/FMO5* (which is not induced by *P. aeruginosa* (Irazoqui et al., 2008, 2010a)). Moreover, *nhr-49/PPARA* RNAi did not abrogate P*fmo-2::gfp* expression in the pharynx (Dasgupta et al., 2020). The reasons of these discrepancies with our observations remain unclear, but could be related to the use of RNAi instead of null alleles, or to pathogen-intrinsic differences between *S. aureus* and *E. faecalis*. Nonetheless, *fmo-2/FMO5* was first described as the most highly induced gene in *E. faecalis*-infected animals compared with *E. coli* controls several years ago (Wong et al., 2007, Yuen and Ausubel 2018). Subsequent work showed that *nhr-49/PPARA* silencing impaired survival of *E. faecalis* infection (Sim and Hibberd, 2016), but the connection between *nhr-49/PPARA* and *fmo-2/FMO5* during infection was not established until our present work and that of others (Dasgupta et al., 2020). Thus, NHR-49/PPAR-α appears to play an important role in defense against a broad range of bacteria, a conclusion that is reinforced by the recent identification of small molecules that protect germline-defective *C. elegans* from *P. aeruginosa*-mediated killing via NHR-49/PPAR-α (Hummell et al., 2021).”

2. The authors should discuss the processes (GO terms) found in their immune vs nutritional response data, and what the two responses share in common, in addition to the nhr-49 mutant data, as this would further increase the insightfulness and impact of this study.

We agree with the Reviewers that better understanding of the overlapping response to fasting and to *S. aureus* infection is of great interest. Alas, the objective of our study was to define the *differences* between the two responses, and thus we compared the fasted animals directly with infected animals by RNA-seq. We did not include a third reference condition (e.g. *E. coli* food) in the experimental setup. This precludes defining the response to fasting and the response to *S. aureus* (with reference to *E. coli* -fed animals) as separate entities that can be compared.

That caveat notwithstanding, we provide analysis of the GO annotations (and other measures of pathway enrichment) for the infection-specific and fasting-specific signatures in Supplementary file 2, which is discussed in L 120-122.

As requested, we provide in Supplementary file 4 the analysis of the *nhr-49* mutant data, compared with wild type, including GO annotations and other measures of pathway enrichment for each of the three gene sets defined by that comparison: NHR-49-dependent, -independent, and compensatory. Supplementary file 4 is discussed in

L 275-337.

3. Similarly, the authors find that NHR-49 is required for fmo-2 induction, but as NHR-49 is a known regulator of fat metabolism in response to starvation, this was not discussed in the context of their findings or the nhr-49 transcriptome data.

We thank the reviewers for identifying this weakness. The comparison of fasted to infected animals in the RNA-seq experiment in Figure 1 and Supplementary file 2 revealed that metabolic functions (e.g. lipid, amino acid, and carbohydrate catabolism) were enriched in the fasted condition but not the infection. For this reason, our discussion of the results centered on the novel and non-metabolic functions of NHR-49. Nonetheless, we recognize that our results could be placed in the general context of known NHR-49 functions better. To address this weakness, we now discuss the context of known NHR-49 metabolic functions in L 526-535.

4. There is no explanation for the use of a 4h time period for post-infection and responses. Since this is a relatively short time, a justification or evidence that this time is appropriate is relevant, as previous studies have looked at longitudinal changes in starvation response, for example.

We based the selection of the 4 h time point on our prior publication (Irazoqui et al., 2010), in which we showed that the “early” response to infection (including *fmo-2*) was already induced by that time.

To clarify this justification, we added the following to the main text: “Moreover, the “early” phase of this response was already upregulated by 4 h infection (Irazoqui et al., 2010a).” L 110-111.

5. In the comprehensive analysis of genes induced by S. aureus infection, only 6 infection-responsive genes were HLH-30-independent and only 2 were NHR-49-dependent. This is very strange, especially since the two genes may share one promoter site (K08C7.4 and K08C7.5), and is not discussed or addressed in the text. How and why would a transcription factor bind only one site in response to a major stress? Similarly is there an effect of K08C7.4 knockdown / knockout? Thus the protective effect of nhr-49(gf) mutants should be diminished / abolished in fmo-2 KO background (and/or K08C7.4).

This is a remarkable insight that escaped us, because we neglected to reflect on the cosmid name for *fmo-2.* We are very grateful to the Reviewer for pointing this out. We specifically addressed this point by: (a) defining the pattern of expression for *K08C7.4* (Figure 3—figure supplement 3A-E), which partially overlaps with that of *fmo-2* but is not induced nearly as highly by infection; (b) defining the effect of *K08C7.4* knockdown in wild type and *nhr-49* loss of function (Figure 9—figure supplement 1A), which showed no effect on infection survival; and (c) defining the effect of *K08C7.4* knockdown in *nhr-49* gain of function (Figure 9—figure supplement 1B), which showed minor reduction of infection survival of only one allele. From these data, we can conclude that *K08C7.4* probably does not play a major role in host defense, with the caveat that RNAi-derived evidence is less robust than that obtained using null alleles (none are available for *K08C7.4*).

To address the point about *fmo-2,* we performed several epistasis experiments (Figure 9). The main conclusion is that *fmo-2* is epistatic to *nhr-49* gain of function both for infection survival and lifespan extension, which supports the model that the main mechanism by which NHR-49 protects against *S. aureus* infection is by inducing *fmo-2.* In addition, there was no biologically significant difference between *nhr-49* and *fmo-2* single and double mutants, which supports the notion that they function in the same pathway.

6. The author's data suggest that fmo-2 expression is not particularly induced in the hlh-30 mutant during S. aureus infection. (Compare Figure 3C and Figure 3D). In contrast, the qRT-PCR data demonstrate that the gene expression levels at this same time have risen >1000 fold in Figure S1 (Note, that in Figure 2D fmo-2 is only ~50-60-fold up. Please, also address this discrepancy). i.e. according to qRT-PCR, fmo-2 expression in the hlh-30 mutant on S. aureus should be similar to infected WT, while according to imaging it's much closer to nhr-49 mutant. This discrepancy between reporter and qRT-PCR needs to be discussed or rectified.

A couple of points to consider: (1) fluorescence microscopy does not possess the same sensitivity and linear range as RT-qPCR, which is the preferred method for expression analysis; (2) the P*fmo-2::gfp* transcriptional reporter is a transgene that only contains 1,536 bp of upstream (promoter) sequence plus the first 16 codons of *fmo-2*, and thus could be missing regulatory elements that are present in the endogenous (chromosomal) locus; (3) steady-state mRNA levels (measured by RT-qPCR) are the net result of the rates of transcription and degradation, and so mRNA stabilization could play a role in increased expression, which would not be reproduced by the GFP reporter. Additionally, gene expression analysis by RT-qPCR can be noisy and can

produce variable results among different trials. (We added similar text in the Materials and methods, L 775-785).

Nonetheless, to mitigate the Reviewer’s concerns, we now include GFP quantification for all induction experiments in the respective figures. In Figure 3K, quantification of GFP restricted to the intestine shows results for *hlh-30* mutants that are more in line with the RT-qPCR (in other words, there is 5-10 fold reduction in GFP in infected *hlh-30* mutants compared with wild type). By volume, the intestine is the largest tissue in the animal (aside from the gonads). Therefore, it is reasonable to expect intestinal expression to drive whole-animal bulk RT-qPCR.

Gene expression analysis can be noisy. To show this point, we plotted the relative expression of *fmo-2* from nine independent biological replicates. *fmo-2* expression in *hlh-30* mutants shows more dispersion than in wild type (Author response image 1). Nonetheless, clearly *fmo-2* expression is defective in *hlh-30* mutants. The magnitude of the defect tends to vary due to reasons we don’t fully understand.

7. Why did authors exclude intestine from the list of tissues where nhr-49-mediated expression of fmo-2 occurs? High magnification images of the FMO-2 reporter strain in hlh-30 and nhr-49 mutants, where intestine-specific fluorescence can be quantified, would be helpful.

L 175-186 now read: “Previous studies identified NHR-49, a nuclear receptor homologous to human PPAR-α and HNF4-α, as essential for *fmo-2/FMO5* induction during exogenous oxidative stress (Goh et al., 2018; Hu et al., 2018). To examine the role of NHR-49/PPAR-α during *S. aureus* infection, we measured *fmo-2/FMO5* expression in *nhr-49/PPARA* null mutants (Liu et al., 1999; Van Gilst et al., 2005). We found that in these mutants, expression of the *fmo-2/FMO5* fluorescent transcriptional reporter was barely induced (Figure 3F, G) and, importantly, was undetectable in the intestinal epithelium (Figure 3H-K). In contrast, *fmo-2/FMO5* induction by *S. aureus* was partially dependent on *hlh-30/TFEB*, as predicted by RNA-seq (Figure 3A-D and G, Figure 2); in these mutants expression was preserved in the pharyngeal isthmus, pharyngeal-intestinal valve, and in the intestinal epithelium, albeit to lower levels compared to wild type (Figure 3H, I, and K).”

Figure 3H-J show representative images of infected animals in the requested backgrounds. Figure 3K shows GFP quantification.